

# Dynamics of ionic species in Svalbard annual snow: the effects of rain event and melting

Elena Barbaro[1], Cristiano Varin[2], Xanthi Pedeli[2,3], Jean Marc Christille[4,5], Torben Kirchgeorg[2], Fabio Giardi[6], David Cappelletti[7], Clara Turetta[1], Andrea Gambaro[2], Andrea Bernagozzi[4], Jean Charles Gallet[8], Mats P. Björkman[9], Andrea Spolaor[1*].

[1]Institute for the Dynamics of Environmental Processes, IDPA-CNR, Via Torino 155, 30172 Venice-Mestre, Italy

[2]Ca' Foscari University of Venice, Department of Environmental Sciences, Informatics and Statistics, Via Torino 155, 30172 Venice-Mestre, Italy.

[3]Athens University of Economics and Business, Department of Statistics, 76 Patision Street, 10434 Athens, Greece.

[4]Astronomical Observatory of the Autonomous Region of the Aosta Valley (OAVdA), Loc. Lignan 39, 11020 Nus (AO), Italy.

[5]Dipartimento di Fisica e Geologia, Università degli Studi di Perugia, I-06123 Perugia, Italy

[6]Chemistry Department – Analytical Chemistry, Scientific Pole, University of Florence, Via della Lastruccia 3, I-50019 Sesto Fiorentino (Florence) Italy.

[7]Dipartimento di Chimica, Biologia e Biotecnologie, Università degli Studi di Perugia I-06123 Perugia, Italy

[8]Norwegian Polar Institute, Tromsø, NO-9296, Norway

[9]University of Gothenburg, Department of Earth Sciences, Box 460, 40530 Göteborg, Sweden

*Correspondence to: Andrea Spolaor IDPA-CNR, (andrea.spolaor@cnr.it)

Keywords: ions, Svalbard, snow, melting effect.





**Abstract.** The Arctic and middle latitude (such as the Alps) ice core archives, except for the Greenland summit, are strongly influenced by melting processes, able to modify the original chemical signal of the annual snowfall. In the last decades, the increase of the average Arctic temperature has caused and enhanced surface snow melting in the higher ice cap, especially in the Svalbard Archipelago. The increase of the frequency and altitude of winter "rain on snow" events as

well as the increase of the length of the melting season has a direct impact on the chemical composition of the seasonal and permanent snow layers due to different migration processes of water-soluble compounds, such as ionic species. The re-allocation along the snowpack of ionic species could significantly modify the original chemical signal present in the annual snow, making comprehensive interpretation of climate records difficult. The chemical composition of the first 100

cm of the seasonal snow at Austre Brøggerbreen Glacier (Spitsbergen, Svalbard Islands, Norway) was monitored daily from the 27$^{th}$ of March until to the 31$^{st}$ of May 2015. The experiment period covers almost the entire Arctic spring until the melting season. During the experiment, a rain event occurred on the 16$^{th}$ to 17$^{th}$ of April while from the 15$^{th}$ of May the snowpack reached an isothermal profile. The presented dataset is unique and helps to better understand the behaviour of cations ($K^+$,

$Ca^{2+}$, $Na^+$, $Mg^{2+}$), anions ($Br^-$, $I^-$, $SO_4^{2-}$, $NO_3^-$, $Cl^-$, MSA) and two carboxylic acids ($C_2$-glycolic and $C_5$-glutaric acids) in the snowpack during this melting period. The results obtained from the experiment give us an overview of how the chemicals are remobilized in the snowpack during a rain event or due to the melting at the end of the spring season. The aim of this paper is to give a picture of the evolution of the seasonal snow strata with the aim to better understand the processes that can

influence the chemical distribution in the annual snow. The results of the present work are unique and helpful for future analyses and interpretation of ice core paleoclimatic archives.



## Introduction

After the first study conducted by Dansgard et al. (1969), ice cores are intensively studied for a better understanding of the climate dynamics (EPICA Community members, 2004). A long list of elements, isotopes and chemical compounds are currently analysed to reconstruct the past climate conditions (Wolff et al., 2010). For examples, calcium, aluminium, iron, and magnesium are studied to quantify the changes in the atmospheric dust load in the past (Wolff et al., 2010). Potassium

could be influenced by multiple sources like sea spray aerosol, dust deposition and biomass burning (Legrand and Mayewski, 1997). Sodium and chloride are used as sea salt tracers to quantify the influences of the marine aerosol (Schüpbach et al., 2018), while stable water isotope ratio reflects atmospheric temperature changes (Stenni et al., 2011). Sulphate and in particular well-defined peaks of non-sea salt sulphate are commonly used to identify volcanic events, supporting the ice

core dating (Sigl et al., 2014). Methanesulphonic acid (MSA) can reflect the past marine primary production and, in a specific case, changes in sea ice conditions (Curran et al., 2003; Isaksson et al., 2005). Parallel to well consolidate climate tracers, other elements and compounds are recently emerging. Bromine and iodine have been suggested to respond to sea ice changes and biological primary production (Cuevas et al., 2018; Spolaor et al., 2016a; Spolaor et al., 2014) and free amino

acids are currently under examination, as possible tracers of marine primary production (Barbaro et al., 2017a).

The seasonal snow layer can be defined as the snow accumulated above the equilibrium line during one year on glaciers while the snow cover that melts during the summer above ground and defined as the snow accumulated and present on the ground during the year. The seasonal snow layer is an

extremely dynamic portion of the cryosphere (Valt and Salvatori, 2016). The snow deposited and accumulated over the glaciers reflects the average atmospheric composition relative to the time of its deposition and its post-depositional processes and preserves information about the transport processes of atmospheric aerosol (Barbante et al., 2004; Spolaor et al., 2014). In the polar region, in particular in the higher and colder ice caps, the annual snow strata can be preserved and the snow

accumulation from year to year then provides proxies for climate reconstructions.

Snow metamorphism is defined as the change of macrophysical snow properties, such as density, grain size, and shape, and it is a function of the temperature gradient within the snowpack (Colbeck, 1982) which is strongly depended on the atmospheric conditions. Several processes can influence the annual signal in the accumulated snow; snow can be removed or restructured by wind; post-

depositional processes can modify the abundance of particular photo-reactive elements; water formation and percolation into the snow pack can redistribute the soluble chemical species present into the snow pack (Brimblecombe et al., 1987). One of the main issues regarding the interpretation





of the chemical signal in ice and snow is the effect of the surface melting and subsequent water percolation, modifying the pristine chemical signal of the annual snow pack and altering the climate information (Pohjola et al., 2002; Vega et al., 2016). Only few studies (Björkman et al., 2014; Eichler et al., 2001; Goto-Azuma et al., 1994; Kuhn, 2001; Pohjola et al., 2002; Vega et al., 2016) have tried to estimate this ion reallocation effects on the annual chemical signals in ice cores, an important issue for ice-climate proxies from regions where summer snow melting occurs, and where will occur more in the next future due to the Arctic temperature rising (Kohler et al., 2007). It is likely that an increasing number of Arctic ice fields and caps, including the Greenland Plateau (Nilsson et al., 2015) will experience summer melting in the near future reducing drastically the location where pristine ice signal could be collected\present.

The results of Pohjola et al. (2002) and Vega et al. (2016) focus on the final effect of the summer melting, describing and characterizing the behaviour of the ionic species into the snowpack\firn. Their results showed an increase of the mobility for ionic species linked with strong acid ($SO_4^{2-}$ and $NO_3^-$). There is still lack of knowledge regarding the fundamental behaviour of ions in the snow pack during the melting phase and, even more important, during specific and rapid meteorological events such as the "rain on snow" events.

The aim of the present research has been to investigate the pristine distribution and the subsequent mobility of soluble species in the snow by monitoring the daily changes of the first meter of the annual snowpack from the early spring (cold climate) until the late spring when the melting phase is on-going. The upper 100 cm of the annual snowpack of the Austre Brøggerbreen glacier (Spitsbergen, Svalbard Archipelago) has been sampled on a daily basis with a resolution of 10 cm from 27[th] March to 31[st] May 2015. Each snow sample was analysed for anions (Br$^-$, I$^-$, NO$_3^-$,SO$_4^{2-}$, Cl$^-$), cations (K$^+$, Na$^+$, Mg$^{2+}$, Ca$^{2+}$), MSA, and two carboxylic acids (C$_2$-glicolic and C$_5$-glutaric acids). The main physical parameters, such as temperature and liquid water content, have been acquired with a continuous measuring system. Temperature sensors (n=11) were positioned with 10 cm depth resolution, while 25 cm depth resolution has been used for the liquid water sensors (n=3). Snow layer identification and hardness test were also performed. This is the first time that such daily sampling has been done in Polar regions and which include a strong melt event, which is unique to study the mobility of the soluble compounds in a changing seasonal Arctic snowpack.



**Experimental design**

**The Sampling site**

The experiment was conducted on the Austre Brøggerbreen (ABB) glacier (78.88916°N, 11.89008°E) in the vicinity of the research facilities in Ny-Ålesund, Svalbard Archipelago. The average annual precipitation in Svalbard ranges from 190 to 525 mm. The Ny-Ålesund area has an annual precipitation of 385 mm and its highest precipitation rates occur in August-October (mainly

rain) and March (mainly snow), while May-June correspond to the lowest rates (Førland et al., 2011). Mean annual temperature is -6.3°C, and February is usually the coldest month (-14.6 ± 3.4 °C) while July is the warmest period (4.9 ± 0.8 °C) (Maturilli et al., 2013). ABB is a small glacier, characterized by a seasonal snow cover that melts during summer (hence, no accumulation region) and has an average snow depth of 2.5 m at the upper part of the glacier. The ABB elevation increase

is constant, with the bottom is located at 45 m a.s.l. and the top is located at 490 m a.s.l., with a length of approximately 7 km and it is confined in a valley by the surrounding mountains. The annual snow pack at the bottom part of the ABB is strongly characterized by melt and refrozen layer while the upper section usually is well preserved with only a few thin melt and refrozen strata (<0.5 cm).

**Sampling strategy**

The snowpack sampling was conducted at the middle of the ABB between the 27th March and 31st May 2015. An area of 10 x 10 m was chosen on the glacier at an elevation of 270 m a.s.l. on a flat area. This area is usually well protected from wind erosion and surface snow redistribution, minimizing post-depositional relocation. Eight plastic poles were used to delimit the designed area

as well as to calculate the accumulation\ablation of the surface snow. Accumulation is needed as a background parameter to identify the evolution of the annual snow pack and correct the snow pit depth for the following sampling. Additionally to the accumulation data, meteorological time series is crucial to understand if the increase in snow depth is due to precipitations or a depletion\erosion a consequence of snowdrift and/or melting. Accumulation data were obtained by daily measurements

of the height of the eight plastic poles compared to the snow surface. A mean value of these measurements was calculated to obtain the average daily loss or gain of snow. The average daily standard deviation of the accumulation measured form the six poles was 3.5 cm.

A one-meter deep snow pit was daily dug perpendicular to the glacier elongation using a clean aluminium shovel. Before sampling the snow, surface was scratch with a dedicated clean

polyethylene shovel. The snow wall was sampled using polyethylene pre-cleaned tubes with a depth increment of 10 cm so that 10 samples were taken each time. After sampling, the snow pit was



carefully filled with the snow previously removed and the surface flattened to avoid wind dune formation and the following day, an adjacent snow pit was dug approximately 30 cm in the upstream direction.

**Snow pit depth correction**

A robust strategy was adopted to compare the 60 snow pits (and the internal snow layer) dug during the experiment, as well as temperature measurements and liquid water content (LWC). Specifically, the strategy considered the changes in the snowpack thickness during the experiment as a consequence of snow deposition or ablation, For the snow sampling depth, the snow surface of the

first snow pit (dug the 27th of March) was considered as a reference (the zero value). The layer depths of the following snow pits have been corrected using the accumulation data collected every day. This correction is necessary to follow correctly the evolution of the chemical data of a given stratum when changes in the snow height may have changed its relative position in the snowpack. In the case of ablation, the difference has been subtracted to the value measured from each pole,

while the difference has been added in case of accumulation. To clarify, the strata initially at 50 cm depth in case of 5 cm ablation will have a depth of 45 cm (subtracting difference), while in case of accumulation of 5 cm will be at 55 cm depth (added difference). The temperature and soil moisture sensors are set up into the snow pack, and their position is also corrected following the same procedure.

**Methods**

**Chemical Analysis**

Determination and quantification of anionic compounds: nitrate ($NO_3^-$), chlorine ($Cl^-$), sulphate ($SO_4^{2-}$), methanesulphonic acid (MSA), and carboxylic acids (glycolic acid (C2) and glutaric acid (C5)) were performed using an ion chromatograph (Thermo Scientific™ Dionex™ ICS-5000,

Waltham, US), coupled with a single mass spectrometer (MSQ Plus™, ThermoScientific™, Bremen, Germany). The separation of anionic compounds was performed with an anion exchange column (Dionex Ion Pac AS11 2x250mm) and guard column (Dionex Ion Pac AG11 2x50 mm) and sodium hydroxide as the mobile phase. The cations: Ammonium ($NH_4^+$), calcium ($Ca^{2+}$), magnesium ($Mg^{2+}$), and potassium ($K^+$) were quantified using the same ion chromatograph also

equipped with a capillary system and a conductivity detector. The separation of cations was carried out using an Ion Pac CS19-4µm capillary cation exchange column (0.4 x250 mm) equipped with an Ion Pac CG19-4µm guard column (0.4x50 mm). All details about both instrumental methods and the detection limits of species are reported in Barbaro et al.(2017a).



Due to the lower concentration of Br and I, the measurements were conducted by Inductively
Coupled Plasma Collision reaction cell mass Spectrometry (CRC-ICP-MSS; Agilent 7500 series)
equipped with a Scott-cooled spray chamber (ESI, Omaha, USA). This analytical system provides
no information about the chemical species but quantifies the total amount of the selected element.
Detection limits, calculated as three times the standard deviation of the blank, were 50 pg g$^{-1}$ for
$^{79}$Br and 10 pg g$^{-1}$ for $^{127}$I. The analytical system was cleaned during 3 min with 2% HNO$_3$ (trace
metal grade, Romil, UK), then 3 min with ultrapure water, and that in cycles during 24h prior each
analyses session. At the beginning of each analysis, a single cleaning cycle was run to return the
background to within 1 % of the initial background level to minimize possible residual memory
effect

**Meteorological data**

The meteorological data have been obtained by the AWIPEV (https://www.awipev.eu/awipev-
observatories) weather station at their research base located in Ny-Ålesund (5 km from the
experimental site) since there are no operating weather stations present on the sampling site. For the
discussion of our results, we considered only the air temperature (PT100 sensor in °C) (Maturilli et
al., 2013). Other meteorological parameters obtained at Ny-Ålesund have not been included in the
discussion because likely influenced by the orography around the ABB glacier. The snow
accumulation has been estimated by measuring the length of six plastic poles placed around the
sampling location while the daily precipitation data were recorded in Ny-Ålesund by the Norwegian
Meteorological Institute (station n. 99910) and downloaded through the eKlima database
(eklima.no). The precipitation data was only included characterizing the precipitation amount
during the rain events since the main important aspect for the experiment is the snow
accumulated\removed to\from the sampling site.

**Temperature and liquid water content**

A continuous system for temperature and LWC, expressed as a percentage of the mass of snow in a
liquid phase, has been installed in the snow at the sampling site. 11 temperature probes were buried
in the snow at 10-cm intervals from the surface down to a depth of 1 m while 3 soil moisture
sensors were buried at 25, 50 and 75 cm depth (depth refers to the first days and corrections have
been applied using the accumulation data). The setup for the thermal and liquid water content
revelation system consists essentially of three parts. The main box contains the electronic
acquisition system, based on Arduino's open-source standard, with a DLS2.0 shield for the logging
part. The second box contains the board for connecting all probes involved in the experiment: 11
temperature probes plus 3 soil moisture sensors. The temperature probes were Maxim Dallas





DS18B20. Their resolution was set to 12 Bits, yielding a readout of up to 750 ms. Pull-up resistors of 4.7 Kohm eliminate all parasite signals. The Liquid water content are derived from soil moisture sensors (SparkFun Soil Moisture Sensor SEN-13322) where the resistance increases as the medium

becomes drier and reaches zero when totally covered by water. The temperature probe was checked by means of one-hour continuous measurements at +25°C and -15 °C controller environment and in parallel with a standard thermometer. The soil moisture sensor for measuring the liquid water content (LWC % per mass) were calibrated by carrying out a measurement in air-dry conditions (0% saturation) and by immersing them in melted snow collected from the sampling site (100 %).

The raw signal value ranged between 0 in absence of liquid water to 680 when fully immersed in melted snow of the sampling site. A more detailed description of the station set up can be found in Spolaor et al. (2016b).

**Snowpack characterization**

The Svalbard snow pack is very variable in term of properties and structure, due to the influence of the ocean and the constant shift between cold, warm, calm and windy conditions. A marked alteration of the internal snow layers structure can occur during rain events, while the snow surface properties will mostly be influenced by solid precipitation and atmospheric conditions (wind, air and radiation). The different snow layers can be distinguished by the density, type, and shape of

snow crystals and the resistance to the penetration (hardness). The properties of a snow layer have a direct effect on the redistribution of ionic species into the snow pack if melting occurs, as it influences the permeability of the snow. During the entire experiment, hardness and stratigraphy were evaluated within 5-days resolution. The resolution was increased in case of specific meteorological events to better understand the evolution of the snow pack layering in concomitance

with strong perturbation.

**Statistical analysis**

The experiment was characterized by two major events, namely the rain on the 16[th] of April and the start of the melting phase defined on the 15[th] of May. The beginning of the melting period was more

difficult to estimate accurately, since some melt-refreezing cycles certainly occurred before the 15[th] of May. Anyway by that time the melting occurred in the entire snowpack, the temperature profile was isothermal and close to the melting point. In these conditions we expect water to be able to percolate from the top down the bottom of the snowpack in one single event without refreezing in the middle of the snow column.



We carried out a statistical analysis to identify which ions were the most significantly affected by each event. All the data were aggregated according to the concentrations levels found in the snow column: three levels corresponding to depths 10-20, 30-50 and 60-90 cm for the biogenic compounds (MSA, $NO_3^-$, C2 and C5) and four levels corresponding to depths 10-40, 50, 60-90 and 100 cm for the other ions. The concentrations were averaged over the depth levels and then

transformed on the logarithm scale to remove the skewness in the distribution of the averaged concentrations.

To investigate if the two events of 16[th] April and 15[th] May had a notable effect on the observed series of each ion, we evaluated the presence of statistically significant local level shifts corresponding to the events. More specifically, we considered observations in a neighborhood of the event and test if the mean level changed after the event. The neighborhood consisted of one

week before and one week after the event. In order to reduce the effect of isolated outliers, the statistical test of a level shift was built upon a linear regression model fitted with a robust M estimator (Huber, 1981), as implemented in the function rlm of the R package (R Core Team, 2018) MASS (Venables and Ripley, 2002). The effect of each event was considered highly significant if

the corresponding p-value is below 0.001, significant if the p-value lies between 0.001 and 0.01 and weakly significant if the p-value lies between 0.01 and 0.05.

**Results**

**Meteorological conditions, snow temperature and liquid water content**

During our experiment, three different synoptic events have been observed, which resulted in three

different temperature gradient regimes in the snow. Therefore, the experiment can be divided into three phases. Phase 1 (March 27 – April 16, P1 in the following) was first characterized by a cold and stable air temperature until a rain event occurred on 17[th] of April. During P1, the temperature gradient in the snowpack was weak (below 10 °C m$^{-1}$) with no melting occurring. Phase 2 (April 17- May 15, P2) was first characterized by the rain event that produced an almost uniform thermal

gradient in the snow, but still with negative temperature values (-3 to -5 °C). A decrease of air temperature followed and cold propagated in the deeper snow layers. Finally, on 15th of May, after a period of warming, the snow temperature profile became almost isotherm but snow surface started to melt, which defines the third period (May 15-30, P3). Air and snow temperature profile together with snow LWC can be seen on Fig.2.

The dry period, P1, the rain event, P2, and the melting period, P3, defined above are in agreement with the LWC data. No liquid water was detected in the P1, while during P2, the LWC increased to 38% over the entire system, which is likely to be overestimated. We hypothesise that the device





created a small confined area where water could be preferentially stored, or that the device got flooded, which explains the very high value recorded, but also the data gap 16 to 18th of April as the electronic system has been damaged by the water. Even after fixing the system, the LWC at 50 cm depth was still as high as 6 to 8% even if the snow temperature decreased down to -10 °C, which is physically unrealistic. We, therefore, do not use the LWC data after that event at 50 cm depth. No liquid water has been detected at 75 cm depth, even during the rain event. During the melting phase, the LWC increased though the entire snow column and finally reached the bottom part of our measured profile at 75 cm on the 26th of May. The constant increase of LWC is also consistent with the propagation of the heat wave through the snowpack.

**Snow structure and layering**

A simplified snow stratigraphy with only the hardness of the snow layers was recorded every 5 days through the experiment (Figure S1). During P1, most of the snow was soft and surface changed occurred due to fresh snow events, wind, compaction of the recent snow and the deeper layers due to snow metamorphosis, which increased the hardness of the top 20 cm of the snow pack. The middle part of the snowpack was medium-hard with some layer up to a hardness index of 4. The bottom of the investigated snow was characterized by a very thick and hard layer, which is typical for Svalbard glacier and consists of the early snow accumulated during late autumn or early winter.

The P2, after the rain event, shows a substantial increase in the hardness of the snowpack as the snow got saturated with water and refrozen (Fig S1). The hardness of the top 20 cm increased with several new ice and melt-refrozen layers and between 20 and 50-60 cm. After the 15th of May, during the melting phase, the hard layers are re-allocated to the bottom of the snowpack. A lot of variability in the appearing and disappearance of the very hard layers is also observed, which is typical for a melting snow pack subjected to refreezing period at night time. A large part of the snow pack became very hard between 40 and 80 cm depth so that a lot of the percolated material got refrozen in these layers. Fresh snow has also been deposited at the surface during that phase.

**Major ions distribution in the snow pack**

The average composition of first 1-m snow was mainly characterized by marine input and the ions related to the sea spray aerosol were the most abundant species, i.e. $Na^+$, $Cl^-$ and $SO_4^{2-}$. The average concentrations ranged between 2 and 8 µg g$^{-1}$ for $Na^+$ (45% of the total sum of all detected ionic species), 1.1 and 3.5 µg g$^{-1}$ for $Cl^-$ (25%), and 0.8 and 3.2 µg g$^{-1}$ for $SO_4^{2-}$ (19%) during the entire experiment (Fig. S2). Magnesium and nitrate represented 4% and 3% of the total sum of all detected ionic species, with concentrations ranged between 0.1 and 0.8 µg g$^{-1}$ and 0.08 and 0.5 µg g$^{-1}$,





respectively. Calcium and potassium were found at the same mean percentage (2%) with concentration values between 81 and 580 ng g$^{-1}$ for Ca$^{2+}$ and 88 and 313 ng g$^{-1}$ for K$^+$. The other ions were found with a percentage below of 1%, as reported in Fig. S2.

For the following discussion, the chemical dataset has been divided into two sub-groups considering both the literature (Legrand and Mayewski, 1997). MSA, NO$_3^-$, C2 and C5 are species
mostly influenced by biogenic activities, such as algal bloom and oceanic primary production (Björkman et al., 2014b; Chebbi and Carlier, 1996; Isaksson et al., 2005). The other species (Na$^+$, Cl$^-$, Ca$^{2+}$, Mg$^{2+}$, K$^+$, Br$^-$) are connected with physical processes such as sea spray aerosol formation and dust transport (long range and local) (Röthlisberger et al., 2002). Sulphate and the total iodine may originate from both sources: sea spray emission and biogenic activities (e.g. algae bloom)
(Maffezzoli et al., 2017; Minikin et al., 1998).

Fig. 2 and 4 report the concentrations of ionic species found in the daily 1-m snow pit with a resolution of 10 cm for the entire experiment (from 27$^{th}$ of March to 31$^{st}$ of May). Considering the profiles reported in Fig. 2 and 4, three different concentration levels can be distinguished for biogenic ions (MSA, NO$_3^-$, C2 and C5) at 10-20 cm, 30-50 cm and 60-100 cm while four
concentration levels can be identified for other ions at 10-40 cm, 50 cm, 60-90 cm and 100 cm. The median concentrations of each ion, divided for each strata and each period, are reported in Tables S1 and S2.

The compounds related with the biogenic emission, in particular MSA, NO$_3^-$, and C5 showed rather homogeneous concentrations until the end of April and the snow layers had low concentrations of
these species without a specific stratification. However slightly higher concentration is detected in the upper stratum. The MSA demonstrated an homogenous concentration in the lower stratum (60-100 cm) with median concentrations ranged between 1 and 2 ng g$^{-1}$ in the three different periods. The superficial layer (10-20 cm) was the most affected stratum and MSA concentrations varied from 16 ng g$^{-1}$ to 34 ng g$^{-1}$ in the P1 and P2, respectively, and 136 ng g$^{-1}$ in the P3. The middle
stratum of MSA was quite stable with median concentrations of 5 ng g$^{-1}$, 14 ng g$^{-1}$ and 22 ng g$^{-1}$ for each consecutive phase, respectively. Similar behaviour was demonstrated for C5, its concentration increased further as a function of time, slowly affecting the deeper layers but never reached the last one (Fig. 2 and Table S1). An increase of the concentration in P3 was also measured for nitrate and C2 but affected almost the entire snow pack, as shown in Fig. 2.

Other compounds (Br$^-$, K$^+$, Mg$^{2+}$, Ca$^{2+}$, Cl$^-$ and Na$^+$) showed a similar distribution during P1 with higher concentration detected in the layers below 50 cm depth and in particular between 50 and 60 cm depth. During the P1, the chemical distribution in the 1-m snow pack remained stable without



any significant changes (Fig. 3). During the P2 and in particular after the rain event, the distribution of the compounds changed, especially for all ions related to sea spray. Saline components showed a

drastic decrease in their abundances. Considering the behaviour of the layer at 50 cm before and after the rain event, the median sodium concentrations varied from 14 µg g$^{-1}$ to 2 µg g$^{-1}$, the concentrations decreased from 1 µg g$^{-1}$ to 0.2 µg g$^{-1}$ for $Mg^{2+}$ and from 4 µg g$^{-1}$ to 2 µg g$^{-1}$ for $Cl^-$ (Table S2). In general, this decrease was observed for all sea spray components (Table S2, Fig. 3), in contrast to nitrate, C2 and C5 (Fig. 2).

The changes in concentrations caused by the rain event remained stable until the second part of the P2 (from the 7$^{th}$ to the 15$^{th}$ of May) where a concentration increase of sulphate and iodine occurred at 50 cm depth. Sulphate and iodine, less impacted by the rain event, presented a concentration increase in the upper stratum from the 5$^{th}$ of May until the end of the experiment. This biogenic input can be confirmed by the time trend of MSA, reported in Fig. 2, and also by evaluating the

temporal profile of non-sea-salt sulfate (nss-$SO_4^{2-}$, Fig. S3). Fig. S3 shows the different profile between $SO_4^{2-}$ and nss-$SO_4^{2-}$, confirming that the biogenic input with the consequent increase of nss-$SO_4^{2-}$ concentrations occurred only in P3 of sampling during the primary production emission. Between the P2 and P3, sulphate showed an increase from 1437 ng g$^{-1}$ to 1713 ng g$^{-1}$ in the upper stratum (10-40 cm) (Table S2), due to the input of primary production. At the contrast, during the

melting phase (P3), the concentration of all sea salt components decreased significantly in whole 1-m snow pack with a tendency to migrate from the upper strata to the bottom part.

Statistical analysis was carried out to identify the effects of the rain event and melting on the chemical structure of the snowpack. Tables 1 and 2 report the summary of p-values of each stratum during the rain event and melting, also indicating the positive and negative level shift. Time series

plots of the averaged concentrations of all ions for each stratum on the logarithm scale are displayed in the Figures S4 and S5 of the supplementary materials.

**Discussion**

During winter 2015, prior to the collection of our data, two rain events occurred: the first one between 22$^{nd}$ and 23$^{rd}$ of January (15.1 mm in total) and the second one the 16$^{th}$ of February (15

mm). These two events impacted an already existing snowpack and formed thick hard layers made with ice, melt-refrozen feature and agglomerated snow crystals between 80 and 100 cm of depth. Such hard strata, especially in presence of ice layers, can act as a physical barrier dividing the annual snow pack in two distinct portions: the lower part representing the period before these events (with possible changes in its chemical signature due to the melt refreezing), and the upper part

which represents the subsequent atmospheric deposition until our sampling.


During P1, concentration values of ionic species are higher in the layers between the 50 to 100 cm depth and in particular at 50 cm depth, where a hard snow layer was observed (Fig. 3). During the same period, sulphate and iodide (Fig. 2) had a similar profile than other ions with the higher concentrations in the lower strata. The two rain events, in January and February, were probably the

main source of these species located in the lower part of the 1-m snow pack. Svalbard rain events are mainly caused by air mass originated from the lower latitudes and able to bring warm and humid air to Svalbard as well compounds accumulated during its travels (Moore, 2016; Rinke et al., 2017). Considering the extended ocean surface southern of Svalbard is likely that these warm air masses are enriched with marine-related elements and compounds that can be precipitated on

Svalbard snow\ice during the rainfall\snowfall. The upper part (0-40 cm) of the snow pack showed lower concentrations and the absence of ice lenses or melt forms in the strata suggests that these deposition events are exclusively made by snowfall, without melting phase. Biogenic organic ions (C5 and C2) showed a weakly increase of concentration in the upper part of snowpack, due to deposition of secondary aerosols (Fig. 2). These organic acids are usually formed by gas-to-particle

conversion via photochemical oxidation of non-methane hydrocarbons and oxygenated organics originated from continental pollution sources and they can be long-range transported in the atmosphere because they are mainly distributed in the fine aerosol particles (Kawamura et al., 2007).

A drastic change in the physical structure and chemical signature of the snow pack occurred in P2,

after the rain event of 16[th] April. From a physical point of view, the presence of liquid water in the snow pack produced the formation of thin ice lenses and refrozen layers in the upper strata (Fig. S1). The net effect of winter\spring rain events on the snow pack has been a rapid warming, with water percolation through the snow column. The superficial strata (0-50 cm of depth) reached temperature near to snow fusion point (Fig. 1), while the lower layers (60-100 cm of depth) showed

temperate always below snow melting point. No water was visible below 75 cm depth because the hard winter layer acted as a barrier and water refreeze on top of it, limiting the percolation.

However, the vertical percolation of the water has the capability to wash out the ions and re-allocate them. Therefore, the change in the distribution of the ions could be a sensitive probe of the percolation process and was evaluated using a statistical approach, defining a negative level shift

when a decrease of concentration occurred while a positive level shift when an improve of concentrations were found.

The degree of statistical significance of the level shifts was summarized with the corresponding p-values and classified into three categories: highly significant if $p < 0.001$, significant if $0.001 \leq p < 0.0$ and weakly significant if $0.01 \leq p < 0.05$. The three categories are commonly adopted in



statistical practice and are used here to avoid overinterpretation of small differences in p-values that are likely statistically irrelevant. Elution sequences were derived accordingly to the three p-value categories. For example, consider four hypothetical ions called A, B, C, D and suppose the p-values for a level shift after the event were $p < 0.001$ for A, $p > 0.05$ for B, $p = 0.003$ for C and $p = 0.007$ for D. The corresponding elution sequence is A > C, D because the level shift is highly significant

for A, significant for B and C, and non-significant for D. Ions C and D are classified at the same degree of significance because we do not want to emphasize small differences in p-values. As a second example, suppose that the p-values for the four hypothetical ions were $p < 0.001$ for A, $p > 0.05$ for B, $p = 0.01$ for C and $p = 0.04$ for D. The corresponding elution sequence is indicated as A >> C, D to indicate the neat difference of significance between the level of shift of A with respect to

those of C and D.

    The stratum at 50 cm of depth was the most influenced by the rain event because all non-biogenic species have a neat negative shift level ($p < 0.001$) except for $Cl^-$ whose shift level is statistically less clear ($p=0.037$) and $Br^-$ with an opposite positive level shift ($p < 0.001$). The main reason for the general decrease in the concentrations after the rain event is linked to the presence of liquid water

with percolation effectively moves ionic compounds to deeper levels (below 100 cm of depth). In fact, this vertical transport affected also the stratum at 60-90 cm depth where some cations showed a negative level shift, following this order of significance $Na^+ > Mg^{2+}, K^+$ (Table 2). The rain event mainly affected the salt species, because biogenic compounds showed higher concentrations in P2 and P3. However, a decrease in the level was detected for $NO_3^-$ ($p < 0.001$), C5 ($p < 0.001$) and

MSA ($p=0.017$) in the 30-50 cm stratum, and for C5 and C2 in the 60-100 cm stratum (Table 1). In contrast, $Br^-$ and $SO_4^{2-}$ showed a positive level shift in the upper stratum (10-40 cm), meaning an increase of concentration after a rain event. This effect is probably due to atmospheric wash-out effect of the wet deposition of the gas phase of bromine and sulphur compounds on the snowpack (Fig. 3, Table 2).

During the experiment, the melting affected substantially the physical and chemical structure of the snowpack. The snow melting started in the upper layer and it propagated into the deeper snow layers with a gradual increase of the LWC (Fig. 1) into the lower snow layer. The ionic species were washed-out at depths of 100 cm, where a hard stratum probably acted as a physical barrier (Fig. S1). The difference between the rain event and the snow melting phase can be identified by

the chemical signature as a consequence of the physical behaviour of wet snow: the rain event produced stronger vertical remobilization while the melting period is horizontally distributed and uniform in the whole snow pack.





The concentration of the majority of ions decreased during the melting due to relocation of ionic species from above to below the 100 cm stratum (Fig. 2 and 4). In contrast to the rain event, the effect of melting started in the upper layers with negative level shifts with this order of significance $Na^+ \gg Cl^-$, $Mg^{2+}$, $Ca^{2+}$ with negative level shifts highly significant for $Na+$ ($p < 0.001$) and weakly significant for $CI-$ ($p=0.012$), $Mg$ ($p=0.03$) and $Ca$ ($p=0.048$). A positive level shift was found for $Br^-$ ($p < 0.001$) in the 10-40 cm stratum and for MSA ($p<0.001$), C5 ($p=0.004$) and C2 ($p=0.013$) in the 10-20 cm. The increase in concentrations of biogenic compounds during the melting period is likely due to a continuous input of these compounds from atmospheric deposition onto the snowpack, where the atmospheric concentrations of these have been linked to marine biological primary production. In contrast, the other species (e.g. saline ions) showed evident effects by the melting, which propagated also in the lower strata (Fig. 5). The stratum at 50 cm of depth showed a negative level shift only for $Ca^{2+}>I^->Mg^{2+}$,$Cl^-$, while all non-biogenic species showed a level shift in the 60-90 cm stratum (Table 2) following this order $Br^-$,$Mg^{2+}$,$K^+$,$SO_4^{2-}$,$I^-$,$Na^+>Ca^{2+}>Cl^-$.

Table 3 summarizes the sequence of preferential elution for each stratum for rain and melting events. The comparison with other previous studies (Table 3) highlighted the complexity to obtain a definitive elution sequence. The first investigation about the elution sequence was published by Johannessen et al. (1977), reporting that 50-80% of the solute species were eluted in the first 30% of melt water (fractionation), following this order: $NH_4^+> Cl^-> Na^+$. Followed this first investigation, several studies were performed in the natural field, highlighted an important deviation from laboratory experiments, models and theories. However, the laboratories investigation (Bales et al., 1989) demonstrated the importance of the initial distribution of chemical species in the pack (mesoscale) and also in the snow grains (microscale) to the timing and relative concentrations of melt water release. Field and laboratory measurements were summarized by Kuhn (2001) reported a preferential elution in the sequence: alkali metals, alkaline earth metals, cations (other than $NH_4^+$) > $SO_4^{2-}>NO_3^->Cl^->NH_4^+> H_2O_2$. Goto Azuma et al. (1994) investigated the same glacier used in this study (ABB) and they estimated the sequence of preferential elution (Table 3) from the ratios of the ionic concentrations at an extraordinary peak observed at the boundary to the average ionic concentrations in the entire top layers above this peak. It was obvious that heavy melting had taken place at the drilling site and that most of the ions had been removed from the layers above to focus in this concentrated peak. Finally, a preferential elution sequence adapted from the modeled meltwater discharge and its measured concentrations were also reported by Björkman et al. (2014) in a glacier close to ABB. This elution sequence (Table 3) differed from the previous studies for Svalbard snow, probably because it reflected an elution sequence from a later stage in the melting process. By this comparison and considering our results, we can conclude that the evaluation of the




elution sequence depends strongly from the stratigraphy of the snowpack and it is very important evaluate each single stratum.

The behaviour of iodine is particularly interesting. Iodine is a chemical species that is reactive under light condition. From a millennial scale, iodine is controlled by atmospheric concentration derived by biological emission (Cuevas et al., 2018). However, this experiment can difficulty assess the role of the primary production since is likely that iodine deposited in the upper snow can be later released back into the atmosphere. The rain event and the snow melting, that cause the presence of liquid water into the snow pack, had a great impact in the iodine distribution in the snow pack (Fig. 2). In particular, the highest concentration of iodine in the snow pack (Table S2) has been detected in the P1 of the experiment. This evidence might be due to the lower incoming solar radiation compare to the later periods, then the rain event produced a significant negative level shift at the 50 cm stratum (Frieß et al., 2010; Spolaor et al., 2014). The melting event produced a negative level shift at the 50 cm and 60-90 cm strata, while the upper stratum (10-40 cm) did not show any statistically relevant level shift probably due to a continuous input for the atmospheric deposition, such as other biogenic markers.

**Conclusions**

A daily sampling of the first meter of snowpack was carried out in the Austre Brøggerbreen glacier (Spitsbergen, Svalbard Islands) from 27th March to 31st May 2015. The main aim of this experiment was to describe the daily physical and chemical changes of a snowpack occurred during a rain event and during the initial period of melting phase is on-going. This is the first field experiment about the evolution of chemical composition of snowpack in a Svalbard glacier. Understanding the dynamics of ions during these specific events is mandatory to use these species as climatic proxies in ice cores archives. A statistical approach was applied to identify the effects of the rain event and the melting phase on the ion re-allocation into the snowpack. This comprehension is becoming important considering the frequency increase of these phenomena in recent years due to changing climate conditions.

The present study describes the drastic change in the chemical structure of the 1-m snowpack produced by rain event because the percolation of liquid water had the capability to wash out the ions and re-allocate them within the snowpack. Saline components ($Na^+$, $Cl^-$, $SO_4^{2-}$, $Ca^{2+}$, $Mg^{2+}$, $K^+$, $I^-$) were mainly affected by the rain events because biogenic species (MSA, C2, C5 and $NO_3^-$) showed lower concentrations in the period when the event occurred. The most influenced stratum was the 50 cm, due to the percolation of liquid water in vertical water channel where the ionic species were washed out in the deeper levels (below of 100 cm). At the contrast, the melting phase



produced a general decrease of all ionic species, starting from the superficial layers, except for the biogenic compounds (MSA, C2 and C5) which showed a positive level shift due to continuous atmospheric deposition. Then, the effect of melting propagated in the lower strata until the hardest strata (ice lens, or melt/refrozen layers) which acted as physical barriers to preserve the allocation of ions.

The present study proposes an unique dataset of ionic composition in the 1-m seasonal snow layer at daily temporal resolution and 10 cm depth resolution. The results obtained suggest significant different dynamics of the ionic compounds during the rain event and during the initial phase of melting. Though ion elution sequences have already been proposed, we demonstrate within this experiment that the elution sequence is not a standardize process and it can change in connection

with different meteorological events, snow physical properties, as well as the liquid water content, complicating the interpretation and the drawing up of an uniform elution sequence. The presented dataset constitutes the first snapshot of the snow pack evolution from chemical and physical point of view based on real data. This study, first of its kind, seeks to improve our comprehension on processes connect with the presence of liquid water in the snow pack, a condition that will become

more frequent in the warming Arctic also in the higher and isolated ice cap.

**Acknowledgments**

We would like to thank all our colleagues at the CNR Dirigibile Italia Arctic Station who worked and helped us during the field campaign. The logistical support of the National Research Council - Department of Earth System Science and Environmental Technologies (CNR-DTA) is gratefully

acknowledged. Special thanks are also due to the Norwegian colleagues and the Norwegian Polar Institute for supporting and joining the field operations. Financial support was also received from the Swedish Foundation of Olle Engkvist Byggmästare to M.P. Björkman. The authors thank Elga Lab water, High Wycombe UK for supplying the pure water systems used in this study.





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





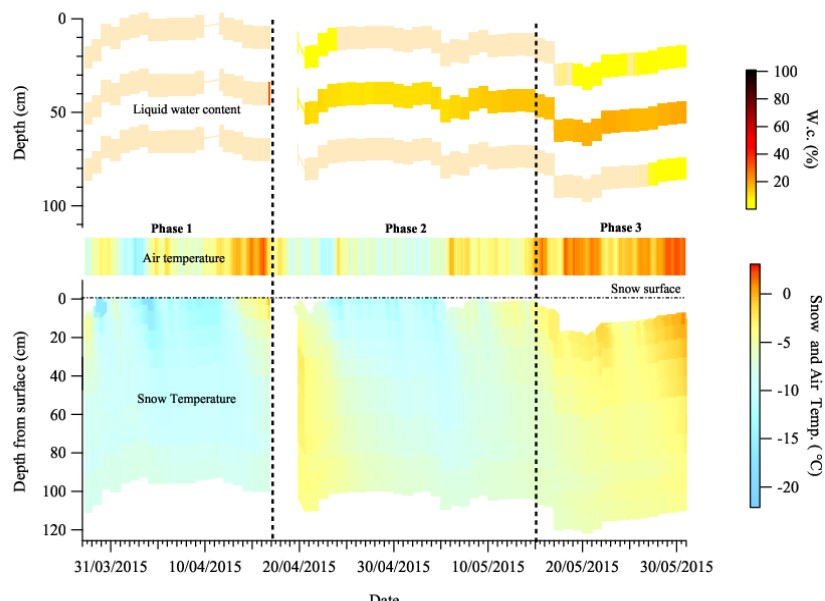

**Fig. 1 a) liquid water content (LWC) measured in three different layers at 25 cm, 50 cm and 75 cm: red indicates the highest LWC value and light blue the lowest (0 % or absence of LWC); b air temperature measured 2 m above the ground at the AWI\IPEV station located in Ny-Ålesund and c) snow temperature of the first meter of snow (corrected in function of the accumulation). Red indicates the warmest (~ +3 °C) and dark blue the coldest (~ -22 °C).**

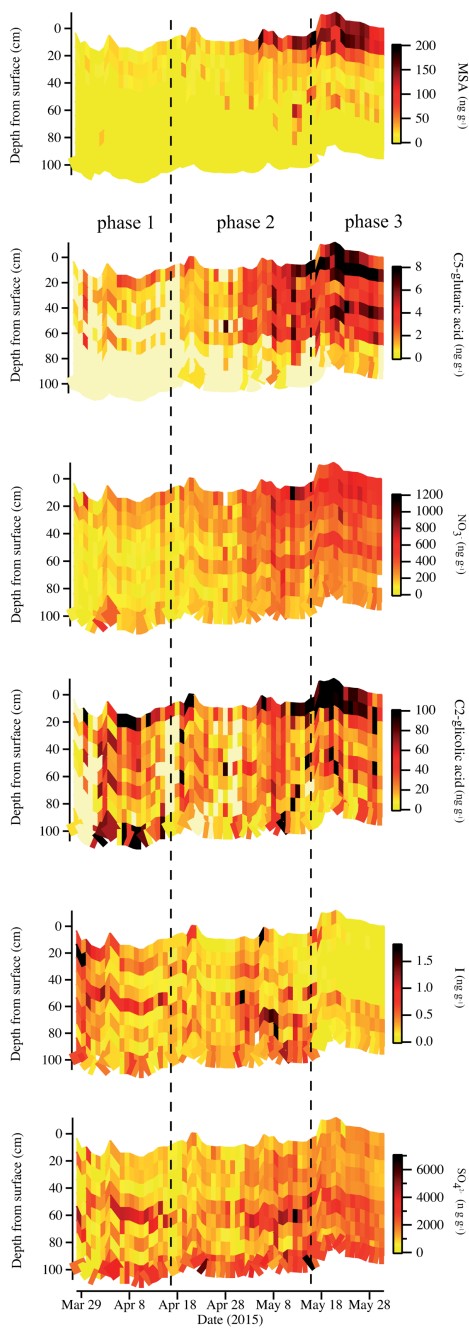


**Figure 2. The concentrations of each biogenic ion (MSA, NO3-, C₅-glycolic and C₅-glutari acids) and total iodide and sulfate (ng g⁻¹) were measured in the daily 1 m snow pit with a 10 cm resolution: dark red represents the highest concentration, light colors the lowest concentration. Each value is corrected for the daily accumulation/ablation.**

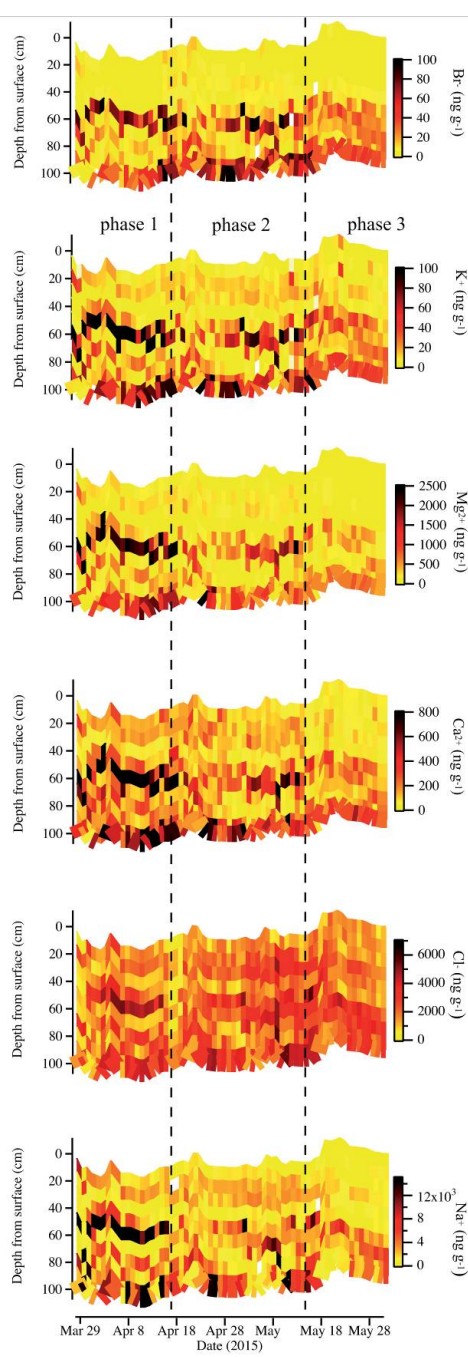


**Figure 3. The concentrations of Br⁻, K⁺, Mg²⁺, Ca²⁺, Cl⁻, and Na⁺ (ng g⁻¹)were measured in the daily 1 m snow pit with a 10 cm resolution: dark red represents the highest concentration, light colors the lowest concentration. Each value is corrected for the daily accumulation/ablation.**





**Table 1.** Summary of the statistical analysis of the biogenic ions series to identify the presence of a positive (+) or negative (-) level shift for each layer after the rain and melting events. P-values below the conventional value of 0.05 are marked in bold. Degree of significance: highly significant $p < 0.001$ ***, significant $0.001 \leq p < 0.01$ **, weakly significant $0.01 \leq p < 0.05$.

| layer | **rain event** | | | **melting event** | | |
|---|---|---|---|---|---|---|
| | *ion* | *Level shift* | *p-value* | *ion* | *Level shift* | *p-value* |
| | NO3 | - | **0.019*** | MSA | + | **<0.001*** |
| 10-20 cm | MSA | - | 0.117 | C5 | + | **0.004** |
| | C5 | - | 0.33 | C2 | + | **0.013*** |
| | C2 | - | 0.805 | NO3 | - | 0.102 |
| | *ion* | *Level shift* | *p-value* | *ion* | *Level shift* | *p-value* |
| | NO3 | - | **<0.001*** | MSA | - | **0.002** |
| 30-50 cm | C5 | - | **<0.001*** | NO3 | - | 0.531 |
| | MSA | - | **0.017*** | C2 | - | 0.588 |
| | C2 | - | 0.202 | C5 | - | 0.616 |
| | *ion* | *Level shift* | *p-value* | *ion* | *Level shift* | *p-value* |
| | C5 | + | **0.004** | C5 | + | **0.002** |
| 60-100 cm | C2 | - | **0.007** | C2 | - | 0.299 |
| | MSA | - | 0.107 | MSA | + | 0.334 |
| | NO3 | - | 0.604 | NO3 | + | 0.942 |






**Table 2.** Summary of the statistical analysis of the non-biogenic ions series to identify the presence of a positive (+) or negative (-) level shift for each layer after the rain and melting events. P-values below the conventional value of 0.05 are marked in bold. Degree of significance: highly significant $p < 0.001$ ***, significant $0.001 \leq p < 0.01$ **, weakly significant $0.01 \leq p < 0.05$.

| layer | rain event | | | melting event | | |
|---|---|---|---|---|---|---|
| | *ion* | *Level shift* | *p-value* | *ion* | *Level shift* | *p-value* |
| | Br | + | **<0.001***** | Na | - | **<0.001***** |
| | SO4 | + | **<0.001***** | Br | + | **0.002**** |
| | K | + | 0.080 | CI | - | **0.012*** |
| 10-40 cm | Ca | + | 0.316 | Mg | - | **0.030*** |
| | Cl | + | 0.399 | Ca | - | **0.048*** |
| | Na | + | 0.473 | SO4 | - | 0.095 |
| | Mg | - | 0.855 | K | - | 0.758 |
| | I | + | 0.961 | I | + | 0.932 |
| | *ion* | *Level shift* | *p-value* | *ion* | *Level shift* | *p-value* |
| | Br | + | **<0.001***** | Ca | - | **<0.001***** |
| | Ca | - | **<0.001***** | I | - | **0.002**** |
| | Mg | - | **<0.001***** | Mg | - | **0.011*** |
| 50 cm | K | - | **<0.001***** | CI | - | **0.026*** |
| | SO4 | - | **<0.001***** | Br | + | 0.057 |
| | I | - | **<0.001***** | Na | - | 0.172 |
| | Na | - | **<0.001***** | SO4 | - | 0.250 |
| | Cl | - | **0.037*** | K | - | 0.393 |
| | *ion* | *Level shift* | *p-value* | *ion* | *Level shift* | *p-value* |
| | Br | + | **<0.001***** | Br | - | **<0.001***** |
| | Na | - | **0.006**** | Mg | - | **<0.001***** |
| | Mg | - | **0.012*** | K | - | **<0.001***** |
| 60-90 cm | K | - | **0.014*** | SO4 | - | **<0.001***** |
| | Ca | - | 0.051 | I | - | **<0.001***** |
| | SO4 | - | 0.131 | Na | - | **<0.001***** |
| | Cl | - | 0.341 | Ca | - | **0.007**** |
| | I | + | 0.797 | Cl | - | **0.010*** |
| | *ion* | *Level shift* | *p-value* | *ion* | *Level shift* | *p-value* |
| | Br | + | **<0.001***** | I | - | **<0.001***** |
| | Mg | - | 0.079 | Ca | - | 0.477 |
| 100 cm | Ca | - | 0.097 | Br | - | 0.709 |
| | Na | - | 0.116 | Na | - | 0.761 |
| | K | - | 0.133 | SO4 | + | 0.833 |



| | | | | | |
|---|---|---|---|---|---|
| SO4 | - | 0.418 | Cl | - | 0.853 |
| I | - | 0.554 | K | + | 0.928 |
| Cl | - | 0.742 | Mg | + | 0.949 |















**Table 3**. Summary of elution sequence obtained in this study for each stratum during rain event and melting phase. A comparison with previous investigations about the preferential elution sequence of melting phase is also reported. In this study the elution sequences are expressed in terms of statistical significance of the level shift following the conventional classification: highly significant $p < 0.001$ ***, significant $0.001 \leq p < 0.01$ **, weakly significant $0.01 \leq p < 0.05$.

| depth | Rain event | Melting phase | |
|---|---|---|---|
| 10-40 cm | $Br^-$***, $SO_4^{2-}$*** | $Na^+$*** >$Br^-$**> $Cl^-$*, $Mg^{2+}$*, $Ca$* | |
| 50 cm | $Br^-$***, $K^+$***, $Mg^{2+}$***, $Ca^{2+}$***, $Na^+$***, $SO_4^{2-}$***, $I^-$*** > $Cl^-$* | $Ca^{2+}$*** > $I^-$** > $Mg^{2+}$*,$Cl^-$* | this study |
| 60-90 cm | $Br^-$*** > $Na^+$** > $Mg^{2+}$*, $K^+$* | $Br^-$***, $K^+$***, $Mg^{2+}$***, $Na^+$***, $SO_4^{2-}$***, $I^-$*** > $Ca^{2+}$**> $Cl^-$* | |
| 100 cm | $Br^-$*** | $I^-$*** | |
| 10-20 cm | $NO_3^-$* | MSA***>C5**>C2* | |
| 30-50 cm | $NO_3^-$***, C2*** > MSA* | MSA** | this study |
| 60-100 cm | C5**, C2** | C5** | |
| Snow pit and firn core at ABB dug in the mid melt season | | $NO_3^-$>$Mg^{2+}$>$SO_4^2$,$Na^+$>$Ca^{2+}$,$Cl$>$K^+$ | (Goto-Azuma et al., 1994) |
| Review of field and laboratory measurements of entire snowpack | | alkali metals, alkaline earth metals, cations (other than $NH_4^+$) > $SO_4^{2-}$>$NO_3^-$ >$Cl^-$ >$NH_4^+$> $H_2O_2$. | (Kuhn, 2001) |
| Snowpit Midtre Lovenbreen, glacier close to ABB | | $Cl^-$, $SO_4^{2-}$> $NO_3^-$, $Na^+$, $K^+$ >$Mg^{2+}$ >$Ca^{2+}$ | (Björkman et al., 2014) |
| Ice core Lomonosovfonna glacier, Svalbard | | $NO_3^-$>$SO_4^{2-}$,$Mg^{2+}$, $Cl^-$, $K^+$, $Na^+$ | (Vega et al., 2016) |