# Peer review of "Dynamics of ionic species in Svalbard annual snow: the effects of rain event and melting"

_The Cryosphere, 2019_

## Referee Comment (RC1) · Anonymous Referee #1 · 16 Jul 2019

General Comments:

This study conducted a series of snowpit chemistry measurements through the spring season in Svalbard with the goal of determining the behavior of snowpack ionic species under the influence of melt and rainfall. The authors present a unique and very comprehensive dataset showing the seasonal-scale evolution of snowpack chemistry in relation to melt and meteorology.

The dataset presented is valuable to our understanding of melt and rain processes and snowpack chemistry and is worthy of being published in a journal such as The Cryosphere. I also commend the authors on the study design and the huge amount of work required to undertake this study.

[Figure]

However, as a manuscript, this paper has several major problems and is not yet suitable for publication. First, the paper contains many errors throughout the manuscript that make the results and the quality of the interpretations very difficult to assess. Figures are referenced incorrectly (or not at all), there are many typos, problems with tense and pluralization and sections of informal or unpolished writing.

While I have confidence in the data themselves, the statistical interpretations of the data were poorly described and were seldom justified by the authors. It is not clear that any of the results or discussion added any insight other than what would be immediately apparent by simply viewing the data. The discussion seems to be an extension of the results, describing the changes in snowpack chemical properties without providing much insight into the mechanisms and processes at play. Aside from a very discrete section on elution sequence, there were insufficient references or context. The few conclusions that the study draws, such as the elution sequence and the effects of the rain events, are not necessarily well-supported by the data (see specific comments).

As it stands currently, this work would be most suitable for a data publication (on the Arctic Data Center, Pangaea or other repositories). Improving the manuscript as it stands would involve bringing in a wider variety of literature – particularly on chemical properties and behavior of ions. I would recommend a more thorough use of meteorological data that potentially includes back trajectory modeling or a comparison of the results with reanalysis products. Many of the problems can be fixed relatively easily with more thorough explanations and clearer writing. I hope that the authors proceed with publishing these results in some form as they represent a tremendous effort and an impressive dataset.

Specific Comments:

I have not made an exhaustive list of mistakes and typos, but some are listed here in addition to specific scientific comments.

Line 26: Also Denali and Logan ice cores have very little/no melt. Consider rephrasing
to "Nearly all ice core archives from the Arctic and middle latitudes, aside from very high elevation sites in Greenland and the North Pacific, are strongly influenced......"

Line 28: "in the higher ice cap" - do you mean at higher elevations on ice caps in general? Or do you mean higher latitudes? Or are you referring to a specific ice cap?

Line 50: Should be "Dansgaard et al." – at least according you your references on line 570.

Line 53: "For example", not "For examples"

Lines 50-65: I think there should be some acknowledgement that while the interpretations listed for each chemical compound listed are commonly used, there is still considerable variation in what each of these proxies mean among sites and a host of factors influencing each of these chemicals.

Line 71: Change to either "over glaciers" or "over the glacier".

Line 74: Higher altitudes or latitudes or both?

Line 78: change to "dependent"

Line 99-111: Very good summary of this project

Lines 106-107: More details are required here. Did you drill a new 100 cm ice core each day and place the ice in a continuous melter system in order to measure physical properties? Why didn't you also sample chemistry if you are going through all this effort? If I am misunderstanding, then more clarification would be helpful.

Lines 118-122: Are these statistics for Ny-Alesund or for your study site? If for Ny-Alesund, how much difference would you except in temperatures at your study site that those you report here, if any?

Lines 143-149: This section is presented well.

Lines 195: It is likely that the orography of the ice cap, the 270 meters of elevation dif-

ference and the albedo differences will also lead to differences in temperature between the station and the field site.

Lines 248-249: Why did you choose these sampling intervals?

Lines 274: You reference Fig. 2, but these data are shown in Fig. 1. Fig. 1: It would be helpful and more quantitative to show the air temperature as a line graph instead of, or in addition to the colorscale displayed in Fig. 1B.

Lines 280-282: If you don't use the data from 50 cm depth after Apr. 18, then it should be removed from Figure 1. Currently it is by far the most striking feature in Fig. 1A and misleading if you think the data are unrealistic because the sensor was broken.

Lines 283-284: You say there was no liquid water was detected at the 75 cm depth, but in the next sentence you say that there was liquid water at 75 cm after May 26. These statements should be reconciled.

Lines 288-302: Since this section essentially describes Fig. S1, I would either move this section to the supplement or move Fig. S1 into the main body of the paper.

Line 314: What do you mean by "considering both the literature"?

Line 321: There is no figure 4. Do you mean Fig. 3?

Line 322-327: These different concentration levels do not stand out to me for most of the plots in Figure 2 and Figure 3 (referenced as Fig.4). How were they chosen? It is also not clear yet, why it is necessary to group data from 10 depth levels into either 3 or 4 depth levels. This is especially true given that the authors just states on lines 318-320 that many of these ions have multiple sources, so it may not make sense to impose differential groupings on the data.

Figs. 2-3: This is a nice way to represent the data. It is a very impressive data set.

Lines 349: To me, it looks like the decrease is only apparent at 50 cm. Is there a statistically significant decrease anywhere else?

Lines 349-350: There is no Table S2. Do you mean Table 2? Or Table 1?

Table 2: Just to clarify: are the values in table 2 calculated from the methods described in lines 252-261? In the figure 2 caption or in the results or discussion, it would be useful to be more explicit than you are when comparing changes in mean concentrations between phases 1, 2 and 3 of different chemicals.

Line 353: What has led you to conclude that the increase in suflate and iodine is biogenic? You haven't given any justification yet.

Line 359: At what contrast? Do you mean "in contrast", or are you talking about the difference between P2 and P3?

Line 360: Do they all decrease significantly? It is not completely clear in Figure 3. What are the significance values and how did you calculate them? Are they in a table?

Line 362: What statistical analysis? How were the p-values calculated? I assume this is using the procedures described in lines 252-261, but you need to remind us. Also, why were these methods chosen? How are they different than a more standard t-test or ANOVA? Have other studies used these methods?

Line 366: What are we supposed to see in these time series plots? I think they are very helpful and probably should be brought to the reader's attention earlier (for instance during the discussion of the shifts between P1, P2 and P3). Figure S4: There are two figures labeled Figure S4. One should be changed to Figure S5.

Line 371: Is this 80-100 cm depth on your normalized depth scale or relative to the surface at the time?

Line 376: This statement does not appear to be true for all of the ionic species. Are you referring to only the ions in Figure 3 but not those in Figure 2? If so, why? And why do some species like chloride not show higher concentrations at depth?

Line 383: Do you mean that there is ice free ocean south of Svalbard? Please correct

this sentence to clarify and fix the grammatical errors.

Lines 379-385: If I understand this section correctly, you argue that that the rain events are from open water areas further to the south, which you infer will have higher ion concentrations. Do you have any data to support this assumption or are there any other studies you can reference that have led you to this conclusion. Since you know the dates of snow and rainfall, and you have been sampling ion concentrations in surface snow, I think you could test this hypothesis with the data in hand. There are a wide variety of tools that can constrain the moisture sources of storms at specific dates. I recommend you further investigate these assumptions using the data you have collected, perhaps combined with back-trajectory modeling to support this discussion.

Line 385-387: It sounds like you are saying that snow = lower concentrations and melt = higher concentrations. This is not necessarily true. Snowfall events can have extremely varied ion concentrations, depending on the source and trajectory of the air mass. And melt can enrich some areas of the snow and dilute other areas. If I have misinterpreted this sentence, then please clarify.

Line 400: Change to "temperature".

Line 404: What statistical approach? Please give details about what you actually did, quantitatively.

Lines 410-420: I'm not sure this is a good way to define the elution sequences among a group of ions. You may have a species with a very low magnitude of elution that is still statistically significant or a high magnitude of elution that is not significant. Just because one ion has a more favorable p value, doesn't necessary mean that that ion elutes first or most strongly. I appreciate the effort involved in sampling snowpits every day, but you do not have a huge number of samples, they are all within a single year and they are affected not just by melt, but many other things as well. Finally, since I don't even know what test you are doing to calculate these p values, I don't know what they mean physically.

Line 423: Why would bromide have a positive level shift? How does this fit with your arguments?

Line 425: How do you know this? It doesn't look like you have made any measurements below 100 cm.

Line 432: Why do you think this? Is this just speculation? Or are there other studies that have shown this? This could be interesting, but you need to back up this claim with a more thorough and fully referenced description of the physical mechanisms you infer.

Line 444: Figure 2 only goes as deep as 100 cm, so how do you know this? There is no Figure 4.

Line 449: Why do you assume a continuous input of biogenic compounds, but not of saline compounds? Why are the saline ions depleted by melt while the biogenic ions appear to be unaffected? What about the chemistry of these ions would be consistent with your data?

Line 456: This paragraph is very full of typos and grammatical errors and is very hard to interpret.

Line 499: What statistical approach?

---

## Referee Comment (RC2) · Anonymous Referee #2 · 24 Jul 2019

General comments

The purpose of this study is to investigate the effects of the rain and snowmelt on ionic species in Svalbard snow. The authors carried out intensive snow-pit observations and snow sampling for about two months on a glacier in Svalbard. They present valuable data obtained from daily snow pit sampling, which requires substantial effort. However, I'm not convinced if their observations, sampling and data analyses were performed adequately to reach their goal. This might be partly due to many typographical/grammatical errors and lack of detailed description, which make some parts of the manuscript hard to understand.

The manuscript gives only qualitative information on different behaviours of biogenic and sea spray ions and elution sequence of ions, without a firm basis.

[Figure]

Without analysing how much melt (and melt/rainwater percolation) caused how much elution of ions, the manuscript doesn't add much to the previous studies. Such quantitative analysis needs calculation of mass balances of H2O and ions in each layer of the snowpack, as was done by e.g. Suzuki (1982, https://www.jstage.jst.go.jp/article/rikusui1931/43/2/43_2_102/_article/-char/en). But the manuscript gives no quantitative information about the amount of fresh snow deposited each day and amount of ions contained in it, nor how much ablation (melting etc.) took place each day. Although the data are unique and important, I don't think the manuscript in its current form provides enough new insight on the effects of the rain and snowmelt for publication in the Cryosphere.

Specific comments

Lines 67-69. I couldn't understand this sentence. I don't understand the definition of the seasonal snow layer. I don't think we usually call the snow layer accumulated above the equilibrium line "seasonal snow layer". What do you mean by "ground"?

Line 69. Does "seasonal snow layer" mean "seasonal snow cover"?

Line 77. I think temperature, as well as temperature gradient, is important.

Line 111. I don't think that snow on a glacier can be called "seasonal snowpack".

Line 118. A reference for the annual precipitation data is necessary here.

Lines 118-122 Where in Ny-Alesund area were the annual precipitation and temperature measured? I think this should be clarified because these values vary with elevation and may vary due to orographic effects.

Line 134. How deep were the plastic poles dug into the glacier ice? Were the poles stable during the whole observation period, not affected by the snowmelt and rain? If the poles moved due to snowmelt or rain, how did the authors correct snow depth readings from the poles?

Lines 141-142. What were the daily loss or gain of snow? I think this information is very important, although the manuscript only gives average daily standard deviations of accumulation.

Lines 144-146. I couldn't understand "Before sampling the snow, surface was scratch". If the surface snow was removed and not collected as a sample, the authors should clarify from what depth (with respect to the snow surface) the samples were taken. I think it is important to collect a sample from the very surface of the snowpack because it is the snow most strongly affected by rain and melting. Furthermore, if new snow deposited after the previous sampling, the newly added amounts of H2O and ions to the snowpack can be measured with this surface sample. This is important to calculate mass balances of H2O and ions. Each day the authors collected only 10 samples from a depth (from the snow surface) which is not written in the manuscript. I'm not convinced if this sampling resolution is enough to trace the temporal changes of same snow layers within the snowpack. To trace the temporal changes of same layers, one should collect samples at a higher resolution or collect samples from the same layer defined by visual stratigraphy (please see Suzuki et al., 1982). I think until heavy snowmelt occurs, one can usually trace a same snow layer by visual stratigraphy of a snowpack. As the thickness of each snow layer within a snowpack changes due to snow densification even if there is no rain events or fresh snow deposition, the authors might have collected snow from different layers on different days, although each sample was collected from a same depth (with depth correction).

Lines 190-193. What is the elevation of AWIPEV. The air temperature at AWIPEV could be different from that at the sampling site. If the manuscript uses the air temperature data at AWIPEV, the temperature difference should be at least estimated.

Lines 199-201. If the daily precipitation was measured near the sea level, it could be different from that at the sampling site. Some part of the rain at the meteorological station could have precipitated as snow at the sampling site at 270 m a.s.l.

[Figure]

Line 232 gives the impression that visual stratigraphy was observed. But later, Line 288 reads that only hardness was recorded. Was visual stratigraphy observed? Visual stratigraphy is very important to deduce the degree of melt effects.

Line 240. Is "close to the melting point" correct? Isn't this "at the melting point"?

Lines 245-261. I couldn't understand how the statistical analysis was performed to investigate the effects of the rain event and snowmelt. More detailed information on statistical analysis is required. Why was the snowpack divided into three levels for the biogenic ions and four levels for other ions? I couldn't see from Figs. 2 and 3. How were p-values calculated and what do they mean?

Line 251. How was the concentration data interpreted using the "logarithmic scale" ?

Line 267. Is the data correct? Isn't it 16th April?

Line 272. Why "but"?

Line 274. Does Fig.2 mean Fig. 1?

Lines 275-286. I'm concerned about the LWC measurement after the rain event. Was the measurement carried out adequately at the depths other than 50cm? Meltwater often percolates through a water channel. If this happens, some part of the snowpack contains liquid water, but other parts at the same depth may not. Have the authors checked the horizontal inhomogeneity in liquid water content? As the authors dug a snow pit every day, they should have observed visually how the meltwater percolated and/or affected the snowpack.

Line 294. How did the authors conclude that this layer consists of the early snow accumulated during late autumn or early winter? Line 124 reads that average snow depth of 2.5 m at the upper part of AAB. What is the average snow depth at the sampling site? If the average snow depth at the sampling site is similar to that at the upper part of AAB ($\sim$2.5m), is the layer at 1 m depth really the snow accumulated in late autumn or early winter? Isn't 1m too shallow?

Line 298. What does "re-allocated" mean?

Line 314. What does "both" mean?

Lines 313-320. Although the authors divided the ions into two groups, I'm not convinced about the grouping. Some of the ions have different sources. For example, nitrate could originate from anthropogenic NOx as well as from biogenic activities. Sulphate could originate from DMS emissions (i.e. biogenic activities), sea spray and anthropogenic SOx. Different sources of an ionic species could have different chemical forms, which could result in a difference in elution sequence.

Line 329. Ions are not compounds.

Lines 328-360. Hard to understand. Descriptions on temporal changes in each ion do not seem to correctly correspond to Figures and Tables. For example, although Lines 328-329 read that MSA showed rather homogeneous concentrations until the end of April, Table S1 shows very different values for three different depth intervals. There are other similar discrepancies between the text and Figures/Tables. The color plots in Figs. 2 and 3 are not easy to understand. Line graphs showing values are necessary here. The authors argue that concentrations of biogenic ions increased in upper layers of the snowpack due to the input of primary production. Supporting evidence is necessary to show that the primary production did increase, air masses with high concentrations of biogenic species did reach AAB, and that these ions did penetrate into the snowpack is necessary. I also wonder why sea spray species did not increase. If air masses from the open sea brought marine biogenic species to AAB, they could have also brought sea spray species. The reason to explain the difference is needed. Why were median concentrations, not averages, used? The authors discuss increases and decreases. Are they statistically significant? There should be spatial variability in ion concentrations in a snowpack. As concentrations were measured in different snow pits on different days, initial spatial variability should be taken into account to discuss temporal variability. I think authors can do that using the daily data from P1.

Line 333. Affected by what?

Lines 363-365. As mentioned above, this part needs more detailed explanation.

Line 369. Where were the values 15.1mm and 15mm recorded? The values would be different at the sampling site.

Lines 387 and 391. Line 387 reads that C5 and C2 are biogenic organic ions, but line 391 reads that these ions originated from continental pollution sources. This is confusing.

Line 400. Does "temperate" mean "temperature"?

Lines 405-406. What does "an improve of concentrations" mean?

Lines 407-420. As stated above, I couldn't understand how the elution sequences were derived.

Lines 421-434. Unless ion concentrations in the rain and fresh snow deposited on the snow surface are known, I think it is difficult to discuss the causes of the changes in ion concentrations.

Lines 440-442. I couldn't understand English here.

Line 453. Fig.5?

Line 480. How is "From a millennial scale" relevant to this study?

Was the density of each snow layer measured? Density measurement is essential for mass balance calculations.

Fig. 1. First, figure title, a, b, and c are missing. Second, the color plots are not easy to understand. In the uppermost plot, value zero is very important, but difficult to see. In the lowermost plot, the melting point is very important, but difficult to see. Line plots showing changes in values are needed here. Third, the manuscript needs to explain how the lowermost plot was made from the 11 temperature probes.

Fig. S1. Visual stratigraphy should be presented. At least new snow and ice layers should be marked.

There are also other typos and errors that I haven't pointed out.

---

## Author Comment (AC1) · 11 Sep 2019

General Comments: R: This study conducted a series of snow pit chemistry measurements through the spring season in Svalbard with the goal of determining the behaviour of snowpack ionic species under the influence of melt and rainfall. The authors present a unique and very comprehensive dataset showing the seasonal-scale evolution of snowpack chemistry in relation to melt and meteorology. The dataset presented is valuable to our understanding of melt and rain processes and snowpack chemistry and is worthy of being published in a journal such as The Cryosphere. I also commend the authors on the study design and the huge amount of work required to undertake this study. However, as a manuscript, this paper has several major problems and is not yet suitable for publication. First, the paper contains many errors throughout the

manuscript that make the results and the quality of the interpretations very difficult to assess. Figures are referenced incorrectly (or not at all), there are many typos, problems with tense and pluralization and sections of informal or unpolished writing.

A: Thanks for all suggestions included in this referee's comment, we modified the manuscript in order to correct several errors and the figure numbers. We completely revised the manuscript.

R: While I have confidence in the data themselves, the statistical interpretations of the data were poorly described and were seldom justified by the authors. It is not clear that any of the results or discussion added any insight other than what would be immediately apparent by simply viewing the data.

A: We expanded the description of the statistical methodology adopted and also added more details in the interpretation of the results. See also our response to your question about the description of the statistical approach that you will find some pages later in this reply letter. Although we agree that a careful and expert view of the data plots may identify most of the effects reported in our analyses, we also think that it is important to quantify the amount of such effects using a proper statistical method. Moreover, there are some doubtful situations where our analyses were used to confirm the significance of the effects of the melting and rain events on the concentrations. We agree with the referee that these results are quite hard to explain, but the understanding of the dynamics of ionic species in snow, especially during specific events, is fundamental. These elements\species are used in paleoclimatic studies but also the release of specific ionic compound, such as nitrate, could have a direct consequence on the vegetative phase of the ground under the snow pack as well microbiological community. Understand the complex process is fundamental for several aspect not only connect with paleoclimate. Papers and research published on this topics try to simply the mechanism evaluating only the macro changes however this experiment bring new light on the complexity of the snow pack system and the chemical diffusion when liquid water Is present. The unique dataset presented will be available for implement snow diffusion models.

R: The discussion seems to be an extension of the results, describing the changes in snowpack chemical properties without providing much insight into the mechanisms and processes at play. Aside from a very discrete section on elution sequence, there were insufficient references or context. The few conclusions that the study draws, such as the elution sequence and the effects of the rain events, are not necessarily well-supported by the data (see specific comments). As it stands currently, this work would be most suitable for a data publication (on the Arctic Data Center, Pangaea or other repositories). Improving the manuscript as it stands would involve bringing in a wider variety of literature – particularly on chemical properties and behavior of ions. I would recommend a more thorough use of meteorological data that potentially includes back trajectory modeling or a comparison of the results with reanalysis products. Many of the problems can be fixed relatively easily with more thorough explanations and clearer writing. I hope that the authors proceed with publishing these results in some form as they represent a tremendous effort and an impressive dataset.

A: The data will be available upon request since the dataset will be too large and complex to be deposit in an on-line data access system with only text explanation.

Specific Comments:

R: I have not made an exhaustive list of mistakes and typos, but some are listed here in addition to specific scientific comments.

A: Thanks to referee, we corrected all suggested mistakes and we checked the entire manuscript to avoid inaccuracy or typos.

R: Line 26: Also Denali and Logan ice cores have very little/no melt. Consider rephrasing to "Nearly all ice core archives from the Arctic and middle latitudes, aside from very high elevation sites in Greenland and the North Pacific, are strongly influenced..."

A: Thanks for the suggestion. We have modified as follows: "Nearly all ice core archives from the Arctic and middle latitudes (such as the Alps), aside from very high elevation

sites in Greenland and the North Pacific, are strongly influenced by melting processes, able to modify the original chemical signal of the annual snowfall."

R: Line 28: "in the higher ice cap" - do you mean at higher elevations on ice caps in general? Or do you mean higher latitudes? Or are you referring to a specific ice cap?

A: As suggested by referee 1, we clarified as follows: "In the last decades, the increase of the average Arctic temperature has caused and enhanced surface snow melting at higher elevations on ice caps, especially in the Svalbard Archipelago".

R: Line 50: Should be "Dansgaard et al." – at least according you your references on line 570.

A: We corrected the mistake.

R: Line 53: "For example", not "For examples"

A: Done

R: Lines 50-65: I think there should be some acknowledgement that while the interpretations listed for each chemical compound listed are commonly used, there is still considerable variation in what each of these proxies mean among sites and a host of factors influencing each of these chemicals.

A: This paper is not focused on the deep discussion of the reliability of the different tracer in ice cores. In the introduction, we want to mention the chemical species, compounds or elements that are commonly used in paleclimate reconstruction. They are used as tracers and not proxy since some aspect of their source, transport and deposition is not always an easy task to decouple. The aim of the paper is to investigate the behaviour of specific ionic species in annual snow pack, to produce a daily picture of their behaviour. Several papers have been published, discussing this argument and giving an elution sequence based on their data. However, our results suggest that the elution sequence is not constant but depends from many parameter linked with the snow physical characteristic. The novelty of our approach is that glaciological

interpretations are driven by statistical quantification of the effects due to the melting and rain events.

R: Line 71: Change to either "over glaciers" or "over the glacier".

A: Done. We used "over glaciers"

R: Line 74: Higher altitudes or latitudes or both?

A: We modified with "higher altitudes".

R: Line 78: change to "dependent"

A: Done

R:Line 99-111: Very good summary of this project

A: Thanks

R: Lines 106-107: More details are required here. Did you drill a new 100 cm ice core each day and place the ice in a continuous melter system in order to measure physical properties? Why didn't you also sample chemistry if you are going through all this effort? If I am misunderstanding, then more clarification would be helpful.

A: All specific details were reported in the "Experimental Section" but, as suggested by the referee, we clarify the experiment as follows: "The upper 100 cm of the annual snowpack of the Austre Brøggerbreen glacier (Spitsbergen, Svalbard Archipelago) was dug daily manually using aluminium shovel from 27th March to 31st May 2015. The exposed snow wall was sampled using polyethylene precleaned tubes with a depth resolution of 10 cm. Then, the snow samples were processed under the laminar flow bench at Dirigibile Italia Arctic Station (Ny-Ålesund) to minimize possible external contamination and transported to Italy for analysis."

R: Lines 118-122: Are these statistics for Ny-Alesund or for your study site? If for Ny-Alesund, how much difference would you except in temperatures at your study site that

those you report here, if any?

A:Refer to next answer.

R: Lines 195: It is likely that the orography of the ice cap, the 270 meters of elevation difference and the albedo differences will also lead to differences in temperature between the station and the field site.

A: We agree with the referee, the measurements derived from the NyA AWS are not fully representative of the temperature at 270 m a.s.l.. Since no operative weather station are available on the glacier and in particular on the sampling site, we used the data available from NyA. The data can be corrected by a factor of 0.65°C\100 m, value often used in meteorology in adiabatic atmospheric condition. To estimate the temperature in our sampling location, the temperature determined in NyA should be decreased of 1.76°C. Correcting the atmospheric data recorded in NyA, the interpretation will not change since the two rain on snow events recorded from January to April are characterized by temperature up to 3\4°C positive. Anyway the text has been improved to be more clear.

R: Lines 143-149: This section is presented well.

A: Thanks

Lines 248-249: Why did you choose these sampling intervals?

A: To clarify how we chose the interval, we modified the manuscript as follows: For the biogenic ions we chosen to split the snow pack in three different layers: 1) the surface layer (10-20 cm) more affected by the wet\dry deposition derived by biological bloom during the period of the experiment; 2) the intermediate level (30-50 cm) to evaluate the possible diffusion of the deposited compounds into the snow pack (until the hard layers detected at 50 cm) and 3) the lower level as background and below the hard layer detect at 50 cm depth. Regarding the other ions we decided to split the snow pack into four layers mainly identify by the snow physical proprieties. The ions, included in this

group, are not directly linked with the biological bloom and can be deposited during the entire snow pack formation. For this reason, the layer division for these species was chosen using the physical snow proprieties at the beginning of the experiment. A layer characterized by soft snow (Hardness index 1 and 2) was detected from 10 to 40 cm depth, and hard layer (Hardness index 4 to 5) at 50 cm depth, and another ice layer at 90 cm depth that divide the lower snow pack into the layer include between 60 and 90 cm depth and at 100 cm depth (Figure 2, discussed later).

R: Lines 274: You reference Fig. 2, but these data are shown in Fig. 1. Fig. 1: It would be helpful and more quantitative to show the air temperature as a line graph instead of, or in addition to the colorscale displayed in Fig. 1B.

A: Thanks to the referee 1. We corrected the mistake. Fig. 1 was modified as suggested and the air temperature is now reported as a line graph. As suggested by referee 2, we modified also the colours to clarify the figure.

R: Lines 280-282: If you don't use the data from 50 cm depth after Apr. 18, then it should be removed from Figure 1. Currently it is by far the most striking feature in Fig. 1A and misleading if you think the data are unrealistic because the sensor was broken.

A: The sensor used to measure the liquid water content are sensors commonly used to measure soil humidity. Is likely that the sensor at 50 cm also after its repairing might have suffered of a bias since the sensor were not calibrate after substitution. However, the sensor seems working since record the enhancement LWC in the snow pack during the melting phase in agreement with the other two calibrated sensors installed at 25 and 75 cm depth. Considering that a -10°C no liquid water can exist, we used the measurements during this day to identify the bias on the sensor and we corrected the dataset data. Thanks to the referee for the precious suggestion.

R: Lines 283-284: You say there was no liquid water was detected at the 75 cm depth, but in the next sentence you say that there was liquid water at 75 cm after May 26. These statements should be reconciled.

A: As suggested by referee, we modified as follows: "No liquid water has been detected at 75 cm depth during the rain event instead, during the melting phase, the LWC increased through the entire snow column and finally reached the bottom part of our measured profile at 75 cm on the 26th of May. The constant increase of LWC is also consistent with the propagation of the heat wave through the snowpack (Fig. 1)."

R: Lines 288-302: Since this section essentially describes Fig. S1, I would either move this section to the supplement or move Fig. S1 into the main body of the paper.

A: As suggested by reviewer 1, we moved the Fig. S1 in the main manuscript. We checked all figure numbers and corrected all mistakes.

R: Line 314: What do you mean by "considering both the literature"?

A: We removed "both".

R: Line 321: There is no figure 4. Do you mean Fig. 3?

A: We checked all figure numbers and corrected all mistakes.

R: Line 322-327: These different concentration levels do not stand out to me for most of the plots in Figure 2 and Figure 3 (referenced as Fig.4). How were they chosen? It is also not clear yet, why it is necessary to group data from 10 depth levels into either 3 or 4 depth levels. This is especially true given that the authors just states on lines 318-320 that many of these ions have multiple sources, so it may not make sense to impose differential groupings on the data.

A: We clarified the main reason of this simplification as follows: "To simplify the huge dataset and to better understand the processes occurred in snow wall, three different concentration levels can be distinguished for biogenic ions (MSA, NO3-, C2 and C5) at 10-20 cm, 30-50 cm and 60-100 cm while four concentration levels can be identified for other ions at 10-40 cm, 50 cm, 60-90 cm and 100 cm. These layers were defined by considering the different ion sources and by evaluating the profiles reported in Fig. 3 and 4."

R: Figs. 2-3: This is a nice way to represent the data. It is a very impressive data set. A: We can confirm that a huge work was carried out to sample, analyze and especially rationalize the dataset.

R: Lines 349: To me, it looks like the decrease is only apparent at 50 cm. Is there a statistically significant decrease anywhere else? Lines 349-350: There is no Table S2. Do you mean Table 2? Or Table 1?

A: Sorry, but we have reported the Table S2 in the Supplementary Material. It repots "Median concentrations of $Br^-$, $K^+$, $Mg^{2+}$, $Ca^{2+}$, $Cl^-$, $Na^+$, $I^-$ and $SO_4^{2-}$ by depth (10-40 cm, 50 cm, 60-90 cm and 100 cm) and phase (I: from 27th March to 16th April; II: from 17th April to 15th May; III: from 16th May to 31st May)." This sentence is included in the "Results" section and, here, we prefer to describe only the concentration results (Table S2). The discussion about the statistically significant changes was reported in the "Discussion" section, where we reported the p-values for each layer. The main conclusion of discussion about the changes in the snow wall is "The most influenced stratum was the 50 cm, due to the percolation of liquid water in vertical water channel where the ionic species were washed out in the deeper levels (below of 100 cm).".

R: Table 2: Just to clarify: are the values in table 2 calculated from the methods described in lines 252-261? In the figure 2 caption or in the results or discussion, it would be useful to be more explicit than you are when comparing changes in mean concentrations between phases 1, 2 and 3 of different chemicals.

A: Yes, the p-values of tables 1 and 2 are calculated as described in the "Statistical analysis". The caption of Fig. 3 (former Fig. 2) has been modified to clarify the different phases.

R: Line 353: What has led you to conclude that the increase in sulfate and iodine is biogenic? You haven't given any justification yet.

A: As suggested by referee, we added one sentence to clarify the biogenic source of

sulfate: "MSA and SO2, which is further oxidized to SO42-, are the main products of dimethyl sulfide (DMS), emitted during algal bloom (Gondwe et al., 2003)." The comparison of temporal trend between MSA and non-sea salt sulphate allowed to confirm the biogenic input of nss-SO42-. Although for iodine other inorganic source might become relevant such as reaction of iodine compound with ozone over the sea water surface (Cuevas et al., 2018) is believed that biological emission are the main source for iodine in the polar areas (Cuevas et al., 2018; Saiz-lopez et al., 2007; Siaz-Lopez et al., 2012).

A: Line 359: At what contrast? Do you mean "in contrast", or are you talking about the difference between P2 and P3?

R: Sorry, we corrected as "In contrast".

A: Line 360: Do they all decrease significantly? It is not completely clear in Figure 3. What are the significance values and how did you calculate them? Are they in a table?

A: The ion concentrations of each phase was reported in the Table S2 of Supplementary Materials and we added this reference in the manuscript. The new description of the statistical analysis provides more details about how the significance values reported in the text and in Tables 1 and 2 are computed, see also our response to the following question.

R: Line 362: What statistical analysis? How were the p-values calculated? I assume this is using the procedures described in lines 252-261, but you need to remind us. Also, why were these methods chosen? How are they different than a more standard t-test or ANOVA? Have other studies used these methods?

A: We are sorry for the quality of presentation. We carefully revised the text to make clear that our discussion is based on the methods described (now with more details) in the section entitled "Statistical Analysis". The statistical analyses discussed in the paper are aimed at the identification of significant change of levels due to the melting

or the rain effects. In the paper was written that the statistical method used to identify such levels changes is "linear regression fitted using a robust M estimator". In fact, this is a simply a t-test with the change of level not estimated using the difference of the sample means before and after the event, but using a robust statistical procedure that is "resistant" to the presence of outliers. The choice of this form of robust t-test in place of the tradition t-test was necessary to avoid incorrect conclusions based on anomalous isolated observations. Statistical methods designed to be resistant to outliers, commonly referenced as "robust statistics", are an important domain of statistics and have been successful applied in a variety of fields. However, we are not aware of any previous use of robust statistical methods in glaciological studies similar to ours. The two academic statisticians who carried out the statistical analyses acknowledge that their description of the statistical methods used in the paper did not suit well the typical audience of the journal. For this reason, the description of the statistical analysis was rewritten adding more details about the employed methodology. We hope that the new description of the statistical methods contains all the elements necessary to evaluate our results.

R: Line 366: What are we supposed to see in these time series plots? I think they are very helpful and probably should be brought to the reader's attention earlier (for instance during the discussion of the shifts between P1, P2 and P3). Figure S4: There are two figures labeled Figure S4. One should be changed to Figure S5.

A: To avoid to create confusion we prefer to leave these figures in the supplementary information. These are the time series plots of averaged concentrations of non-biogenic and biogenic ions, that are produced to apply the statistical approach.

R: Line 371: Is this 80-100 cm depth on your normalized depth scale or relative to the surface at the time?

A: To clarify, we added "from the surface" after "between 80 and 100 cm of depth".

R: Line 376: This statement does not appear to be true for all of the ionic species. Are

you referring to only the ions in Figure 3 but not those in Figure 2? If so, why? And why do some species like chloride not show higher concentrations at depth?

A: As suggested by referee #1, we modified "ionic species" with "sea salt ions" reported in Figure 4. Moreover, we added an explanation about the Cl- behavior: "In contrast, chloride demonstrated a homogenous concentration through the snow strata, because chloride depletion can occur with the HClgas formation, with consequent mobilization in porous snow strata (De Angelis and Legrand, 1995)."

R: Line 383: Do you mean that there is ice free ocean south of Svalbard? Please correct this sentence to clarify and fix the grammatical errors.

A: We tried to clarify the concept as follows: "During the same period, sulphate and iodide (Fig. 3) had a similar profile than sea salt ions with the higher concentrations in the lower strata. These high concentrations can be due to the deposition by two rain events occurred in January and February of 2015. These rain events are mainly caused by warm air mass enriched with marine-related elements originated in the ice free ocean surface in southern of Svalbard (Moore, 2016; Rinke et al., 2017)."

R: Lines 379-385: If I understand this section correctly, you argue that that the rain events are from open water areas further to the south, which you infer will have higher ion concentrations. Do you have any data to support this assumption or are there any other studies you can reference that have led you to this conclusion. Since you know the dates of snow and rainfall, and you have been sampling ion concentrations in surface snow, I think you could test this hypothesis with the data in hand. There are a wide variety of tools that can constrain the moisture sources of storms at specific dates. I recommend you further investigate these assumptions using the data you have collected, perhaps combined with back-trajectory modeling to support this discussion.

A: As suggested by referee, we added to the Supplementary material the Figure S6, which reported the back-trajectories related to the two rain events of January and February 2015. To confirm our hypothesis, we modified the paragraph in the

manuscript as follows: "These high concentrations can be due to the deposition by two rain events occurred in January and February of 2015. These rain events are mainly caused by warm air mass enriched with marine-related elements originated in the ice free ocean surface in southern of Svalbard (Moore, 2016; Rinke et al., 2017), as demonstrate by back-trajectories calculated for these two rain events (Fig. S6)"

R: Line 385-387: It sounds like you are saying that snow = lower concentrations and melt = higher concentrations. This is not necessarily true. Snowfall events can have extremely varied ion concentrations, depending on the source and trajectory of the air mass. And melt can enrich some areas of the snow and dilute other areas. If I have misinterpreted this sentence, then please clarify.

A: This sentence was connected with the previous section where we discussed the presence of high concentration in the lower strata. To avoid confusion, we prefer to remove it.

R: Line 400: Change to "temperature".

A: Done

R: Line 404: What statistical approach? Please give details about what you actually did, quantitatively.

A: We are sorry for the quality of presentation. We carefully revised the text to make clear that our discussion is based on the methods described (now with more details) in the section entitled "Statistical Analysis".

R: Lines 410-420: I'm not sure this is a good way to define the elution sequences among a group of ions. You may have a species with a very low magnitude of elution that is still statistically significant or a high magnitude of elution that is not significant. Just because one ion has a more favorable p value, doesn't necessary mean that that ion elutes first or most strongly. I appreciate the effort involved in sampling snowpits every day, but you do not have a huge number of samples, they are all within a single

year and they are affected not just by melt, but many other things as well. Finally, since I don't even know what test you are doing to calculate these p values, I don't know what they mean physically.

A: As explained in the revised section "Statistical Analysis" we define the elution sentences on the basis of the t statistics that measure the "standardized" amount of change in the week subsequent to melting or rain event. The standardization implicit in the t-statistics is needed to compare ions measured on quite different scales. The p-values reported in the paper are in one-to-one correspondence with the absolute values of the t statistics and provide an alternative way to summarize the effects of the two events in terms of statistical significance. Since the amount of data is limited and we want to avoid over-interpretation of small differences in the t-statistics that are likely insignificant, then we followed the standard practice in statistics of declaring effects as highly significant, significant or weakly significant accordingly to different levels of the p-values.

R: Line 423: Why would bromide have a positive level shift? How does this fit with your arguments?

A: During spring time, Br can be also emitted from the sea ice as BrO through the mechanism of bromine explosion. Bromine explosion over sea ice can lead to an enrichment of Bromine compare to the sea water abundance in the snow deposition causing and enhancement of Br in the snow pack. For example, in the Greenland plateau the Br can be 20 times higher compared to the sea water abundance. It is likely that during the rain event part of the gas phase bromine present over Svalbard have deposited into the snow pack causing the increase of concentration.

R: Line 425: How do you know this? It doesn't look like you have made any measurements below 100 cm.

A: This is our hypothesis but further investigations are needed to confirm it. For this reason, we modified the manuscript as follows: "The main reason for the general decrease in the concentrations after the rain event may be linked to the presence of liquid water with percolation of ionic compounds to deeper levels (below 100 cm of depth). Further investigations are necessary to confirm this hypothesis."

R: Line 432: Why do you think this? Is this just speculation? Or are there other studies that have shown this? This could be interesting, but you need to back up this claim with a more thorough and fully referenced description of the physical mechanisms you infer.

A: "Br- and SO42- showed a positive level shift in the upper stratum (10-40 cm), meaning an increase of concentrations after a rain event. For Br-, this effect is probably due to atmospheric wash-out effect (Spolaor et al., 2019) of the wet deposition of the gas-phase of bromine emitted from sea ice during spring around Svalbard. Regarding sulphur compounds, an extra source of nss-SO4 might undergo a transport from lower latitude (Fig. S3) causing a slight sulphate increase during the rain events in upper snow pack (Fig. 3 and 4, Table 2)."

R: Line 444: Figure 2 only goes as deep as 100 cm, so how do you know this? There is no Figure 4.

A: We agree with referee#1 and some mistakes about the figure numbering created confusion. We corrected the figure numbers and we introduce a "probable" because we hypothesized that "The concentration of the majority of ions decreased during the melting due to probable relocation of ionic species from above to below the 100 cm stratum (Fig. 3 and 4)."

R: Line 449: Why do you assume a continuous input of biogenic compounds, but not of saline compounds? Why are the saline ions depleted by melt while the biogenic ions appear to be unaffected? What about the chemistry of these ions would be consistent with your data?

A: As reported in the manuscript, we assume that the increase the biogenic compounds concentration in May (during the melting period) was likely due to the input of atmospheric deposition because in this period the marine biological primary production occurred and the emission of these species from the sea improved. To clarify the concept, we modified the manuscript as follows: "The increase in concentrations of biogenic compounds during the melting period is likely due to a continuous input of these compounds from atmospheric deposition onto the snowpack, because marine biological primary production increased in this period with the consequent increase in the emission of these compounds."

R: Line 456: This paragraph is very full of typos and grammatical errors and is very hard to interpret.

A: Sorry for the mistakes, we corrected the paragraph as follows: "Table 3 summarizes the sequence of preferential elution for each stratum related to rain and melting events. The comparison with previous studies (Table 3) highlighted the complexity of the main goal of this research. The first investigation about the elution sequence was published by Johannessen et al. (1977), and they reported that 50-80% of the solute species were eluted in the first 30% of melt water (fractionation), following this order: $NH4+>$ $Cl-> Na+$. Several other studies were performed in the natural field, highlighted an important deviation from laboratory experiments, models and theories."

R: Line 499: What statistical approach?

A: We are sorry for the quality of presentation. We revise the text of the conclusion to make clear that the conclusions are based on the methods described in the section entitled "Statistical Analysis".

---

## Author Comment (AC2) · 11 Sep 2019

General comments

The purpose of this study is to investigate the effects of the rain and snowmelt on ionic species in Svalbard snow. The authors carried out intensive snow-pit observations and snow sampling for about two months on a glacier in Svalbard. They present valuable data obtained from daily snow pit sampling, which requires substantial effort. However, I'm not convinced if their observations, sampling and data analyses were performed adequately to reach their goal. This might be partly due to many typographical/ grammatical errors and lack of detailed description, which make some parts of the manuscript hard to understand.

[Figure]

A: We completely the entire manuscript to correct the typographical/ grammatical errors. We clarify parts of the manuscript as suggest by both referees.

R:The manuscript gives only qualitative information on different behaviours of biogenic and sea spray ions and elution sequence of ions, without a firm basis. Without analysing how much melt (and melt/rainwater percolation) caused how much elution of ions, the manuscript doesn't add much to the previous studies.

A: The statistical approach used in the manuscript has the main aim to define quantitatively evaluate the effect of specific events (rain on snow and melting periods) in the elution sequence of ions. This is the first time that a robust multi-layer statistical approach is used, allowing to define if the changes occurred to the ionic composition and give an overview of a possible elution sequence. The results highlight that the elution sequence is strongly dependent by different parameter (most likely linked to the snow physic) and a unique elution sequence could be difficult to define.

R: Such quantitative analysis needs calculation of mass balances of H2O and ions in each layer of the snowpack, as was done by e.g. Suzuki (1982, https://www.jstage.jst.go.jp/article/rikusui1931/43/2/43_2_102/_article/-char/en). But the manuscript gives no quantitative information about the amount of fresh snow deposited each day and amount of ions contained in it, nor how much ablation (melting etc.) took place each day. Although the data are unique and important, I don't think the manuscript in its current form provides enough new insight on the effects of the rain and snowmelt for publication in the Cryosphere.

A: Fresh snow deposited were evaluated each day and for this reason and present in the section "Snow pit depth correction". The accumulation data, different compare the snow deposition, were obtained by daily measurements of the height of the eight plastic poles compared to the snow surface. A mean value of these measurements was calculated to obtain the average daily loss or gain of snow. The average daily standard deviation of the accumulation measured form the six poles was 3.5 cm. We

must note that we are evaluating the accumulation over the snow pack after a snow fall instead of total snow fall deposition (instrumental measured), since part of the snow fall can be removed by wind blow.

Specific comments

R: Lines 67-69. I couldn't understand this sentence. I don't understand the definition of the seasonal snow layer. I don't think we usually call the snow layer accumulated above the equilibrium line "seasonal snow layer". What do you mean by "ground"? Line 69. Does "seasonal snow layer" mean "seasonal snow cover"?

A: We clarified the sentence as follows: "The seasonal snow layer can be defined as the snow accumulated and present on the ground (including glacier surface) during the year and melts during the summer."

R: Line 77. I think temperature, as well as temperature gradient, is important.

A: As suggested by referee#2, we modified as follows: "Snow metamorphism is defined as the change of macrophysical snow properties, such as density, grain size, and shape, and it is a function of temperature, as well as temperature gradient within the snowpack (Colbeck, 1982)"

R: Line 111. I don't think that snow on a glacier can be called "seasonal snowpack".

A: The snow accumulates during one year on a Svalbard glacier is transformed, above the equilibrium line, into firn (density above 500 kg m-3) during the summer period. The snow firn transition is clear, this is the reason why we can define the seasonal snowpack also above glacier surface. In the glacier below the equilibrium line, the snow deposited along one year is completely removed\melted. In both cases is possible distinguished the seasonal snow pack.

R: Line 118. A reference for the annual precipitation data is necessary here.

A: We obtained the annual precipitation data through the eKlima database (eklima.no).

[Figure]

R: Lines 118-122 Where in Ny-Alesund area were the annual precipitation and temperature measured? I think this should be clarified because these values vary with elevation and may vary due to orographic effects.

A: We agree with the referee, the measurements derived from the NyA AWS are not fully representative of the temperature at 270 m a.s.l. Since no operative weather station is available on the glacier and in particular on the sampling site, we can adopt and correct the temperature of 0.65°C\100 m, a value often used in meteorology. To estimate the temperature in our sampling location, the temperature determined in NyA decreased of 1.76°C. The new dataset obtained will not affect the interpretation since the two "rain on snow" events recorded from January to April are characterized by temperature up to 3\4°C positive. The text in the manuscript has been revised as also suggested by the other referee: "Air temperature is measured at sea level while our sampling site is located at 270 m a.s.l. Negative differences between our site and Ny-Alesund are expected considering the temperature decreasing of 0.65°C\100 m in adiabatic atmospheric condition (the temperature measured NyA should be correct for 1.76°C). However, this is meteorological approximation in the free atmosphere and this calculation is used in a synoptic condition affect. For this reason, the surrounding orography could lead to over- or under-estimation of value."

R: Line 134. How deep were the plastic poles dug into the glacier ice? Were the poles stable during the whole observation period, not affected by the snowmelt and rain? If the poles moved due to snowmelt or rain, how did the authors correct snow depth readings from the poles?

A: The poles were 2.40 meters long and dug of one meter into the snow pack until the hard layers determined at 100 cm depth. In addition the humidity sensor install at 75 cm never measured liquid water content until the end of the experiment (as well the temperature never rise above -3°C ) suggesting the stability of this layer where the poles was settled. R: Lines 141-142. What were the daily loss or gain of snow? I think this information is very important, although the manuscript only gives average daily

standard deviations of accumulation.

A: The daily loss\gain of the snow is simply summarized on the snow elevation in all figures.

R: Lines 144-146. I couldn't understand "Before sampling the snow, surface was scratch". If the surface snow was removed and not collected as a sample, the authors should clarify from what depth (with respect to the snow surface) the samples were taken. I think it is important to collect a sample from the very surface of the snowpack because it is the snow most strongly affected by rain and melting. Furthermore, if new snow deposited after the previous sampling, the newly added amounts of H2O and ions to the snowpack can be measured with this surface sample. This is important to calculate mass balances of H2O and ions.

A: We apologize if the text was not clear enough. What we meant was different as explained in the revised paper: "Before sampling, the snow wall was cleaned, removing the first stratum with a dedicated clean polyethylene shovel in order to prevent a possible contamination by the aluminum shovel."

R: Each day the authors collected only 10 samples from a depth (from the snow surface) which is not written in the manuscript.

A: We reported this sentence in the manuscript: "The snow wall was sampled using polyethylene pre-cleaned tubes with a depth increment of 10 cm so that 10 samples were taken each time."

R: I'm not convinced if this sampling resolution is enough to trace the temporal changes of same snow layers within the snowpack. To trace the temporal changes of same layers, one should collect samples at a higher resolution or collect samples from the same layer defined by visual stratigraphy (please see Suzuki et al., 1982). I think until heavy snowmelt occurs, one can usually trace a same snow layer by visual stratigraphy of a snowpack. As the thickness of each snow layer within a snowpack changes due to

snow densification even if there is no rain events or fresh snow deposition, the authors might have collected snow from different layers on different days, although each sample was collected from a same depth (with depth correction).

A: The investigation of the same snow strata for two months is not easy since some strata might completely transformed during warm events (see the stratigraphy in figure 2). Both the two strategies suggested by the reviewer have already been adopted leading to similar results. Considering the enormous amount of the snow pit dug we preferred to adopt a constant depth interval for two reasons: 1) use a standardized method that allows to avoid different operation-specific interpretations, and 2) simplify the data comparison among different days. The aim of the paper is not to follow the strata but to investigate the concentration and its changes of the upper 100 cm of the annual snow pack; there are different opinions about the presence of specific elements in particular snow layers and for sure this could be an interesting topic to further investigate.

R: Lines 190-193. What is the elevation of AWIPEV. The air temperature at AWIPEV could be different from that at the sampling site. If the manuscript uses the air temperature data at AWIPEV, the temperature difference should be at least estimated.

A: See reply for comments line 118-122

R: Lines 199-201. If the daily precipitation was measured near the sea level, it could be different from that at the sampling site. Some part of the rain at the meteorological station could have precipitated as snow at the sampling site at 270 m a.s.l.

A: We agree with the referee, but the temperature during the rain events in Ny-Alesund reached the 4°C. Considering the temperature decrease with altitude (0.65°C\100 m), the temperature at the sampling site might be colder of 1.75 °C, this is true also during the melting phase. However this value are used in the free troposphere and might not be representative for site where the orography cannot be excluded and might increase the atmospheric mixing efficiency. Anyway during the rain

events if the 16th of April the system for snow temperature and LWC was burn due to water infiltration. Temperature data are also available at the Zeppelin station (http://ebas.nilu.no/Pages/Plot.aspx?key=DBCBA03A77D54265868218D4E5E63521). This station is located at 475 m a.s.l. During the 16th of April 2015 temperature reached 0°C.

R: Line 232 gives the impression that visual stratigraphy was observed. But later, Line 288 reads that only hardness was recorded. Was visual stratigraphy observed? Visual stratigraphy is very important to deduce the degree of melt effects.

A: As also suggested by referee#1, we added the Figure 2 in the main manuscript to show the evolution of the snow stratigraphy in the first meter of annual snow.

R: Line 240. Is "close to the melting point" correct? Isn't this "at the melting point"?

A: Thanks. We correct "at the melting point"

R: Lines 245-261. I couldn't understand how the statistical analysis was performed to investigate the effects of the rain event and snowmelt. More detailed information on statistical analysis is required.

A: The statistical analyses discussed in the paper are aimed at the identification of significant change of levels due to the melting or the rain effects. In the paper was written that the statistical method used to identify such change of levels is "linear regression fitted using a robust M estimator". In fact, this is a simply a t-test with the change of level not estimated using the difference of the sample means before and after the event, but using a robust statistical procedure that is "resistant" to the presence of outliers. The choice of this form of robust t-test in place of the tradition t-test was necessary to avoid incorrect conclusions based on anomalous isolated observations. The two academic statisticians who carried out the statistical analyses acknowledge that their description of the statistical methods used in the paper did not suit well the typical audience of the journal. For this reason, the description of the statistical analysis was completely

rewritten adding more details about the employed methodology. We hope that the new description of the statistical methods contains all the elements necessary to evaluate our results.

R: Why was the snowpack divided into three levels for the biogenic ions and four levels for other ions? I couldn't see from Figs. 2 and 3.

A: As also suggested by referee #1, we clarified the main reason of this simplification as follows: "To simplify the huge dataset and to better understand the processes occurred in snow wall, three different concentration levels can be distinguished for biogenic ions (MSA, NO3-, C2 and C5) at 10-20 cm, 30-50 cm and 60-100 cm while four concentration levels can be identified for other ions at 10-40 cm, 50 cm, 60-90 cm and 100 cm. These layers were defined by considering the different ion sources and by evaluating the profiles reported in Fig. 3 and 4."

R: How were p-values calculated and what do they mean?

A: As explained in the revised description of the statistical analyses, the p-values are computed from robust t statistics and they identify whether the average levels of the concentrations change significantly during the week subsequent the melting or rain event.

R: Line 251. How was the concentration data interpreted using the "logarithmic scale"?

A: The logarithm transformation was necessary for the optimality of the statistical analyses. Since the logarithm is a monotonic increasing transformation, then it does not affect the interpretation of the results: that is identification of significant "jumps" in the concentrations after the events.

R: Line 267. Is the data correct? Isn't it 16th April?

A: Thanks, we corrected the mistake.

R: Line 272. Why "but"?

[Figure]

A: We removed "but still".

R: Line 274. Does Fig.2 mean Fig. 1?

A: Thanks for noticing the wrong figure numbering, now fixed in the revised paper.

R: Lines 275-286. I'm concerned about the LWC measurement after the rain event. Was the measurement carried out adequately at the depths other than 50cm? Meltwater often percolates through a water channel. If this happens, some part of the snowpack contains liquid water, but other parts at the same depth may not. Have the authors checked the horizontal inhomogeneity in liquid water content? As the authors dug a snow pit every day, they should have observed visually how the meltwater percolated and/or affected the snowpack.

A: The sensor was located (and later stuck to snow settles) into the snow and was not possible to carry out a spatial variability with the instrument. Moving the instrument for temperature and LWC measurements meant to dig several snow pit and to modify the original structure of the snow pack. The measure of the spatial variability of the LWC was possible only with more than one device set up on the glacier or using more sensors located at the same depth. The device used was calibrated, also in term of energy consumption to work with the number of sensors described in the manuscript. We agree with the referee that more sensors are necessary to better check the spatial variability of LWC, a parameter that is quite complex to measure in the snow pack. The horizontal homogeneity was checked every day during the snow chemical sampling. The snow pit was characterized by a sampling wall of 1.5 m wide where the main features (i.e. ice layer, melt refrozen strata) were evaluated. The main features showed a good homogeneity, although it must be note that the hard layers (melt and refrozen layer) at 50 cm and 100 cm were used as reference since their homogeneous distribution in the snow pack was investigated. Some thin layers produced by the rain event as well by the temperature rising at the end of experiment (in general <0.5 cm thick) were inhomogeneous. As wrote in the manuscript, we know that preferential percolation in specific channels is possible, however the LWC device was used to determine a change in the liquid content and not the absolute value as remark in the manuscript.

R: Line 294. How did the authors conclude that this layer consists of the early snow accumulated during late autumn or early winter? Line 124 reads that average snow depth of 2.5 m at the upper part of AAB. What is the average snow depth at the sampling site? If the average snow depth at the sampling site is similar to that at the upper part of AAB (_2.5m), is the layer at 1 m depth really the snow accumulated in late autumn or early winter? Isn't 1m too shallow?

A: Two heavy rain events occur during 2015. The first one on January 22nd and 23rd and the second the 16th of February. Is not always easy to date a snow pit since wind blow, sublimation ect can modify the pristine snow pack. However, the rain events left a clear sign into the snow pack. Most likely the melt refrozen strata at 50 cm depth was caused by the events occurred the 16th of February while the hard strata at 100 cm depth was caused by the events occurred in January. Since only two layers in the snow pack have been detected, we can have a roughly estimation of the snow pack aged. The full depth of the snow pack was 1.65 m. In general the higher accumulation of snow occurred between January and April (Spolaor et al., 2016).

R:Line 298. What does "re-allocated" mean?

A: As suggested by referee, we modified as follows: "the hard layers shifted to the bottom of the snowpack"

R: Line 314. What does "both" mean?

A: We removed "both".

R: Lines 313-320. Although the authors divided the ions into two groups, I'm not convinced about the grouping. Some of the ions have different sources. For example, nitrate could originate from anthropogenic NOx as well as from biogenic activities. Sulphate could originate from DMS emissions (i.e. biogenic activities), sea spray and anthropogenic SOx. Different sources of an ionic species could have different chemical forms, which could result in a difference in elution sequence.

A: We tried to simplify the huge dataset, considering the profile reported in the Figures 3 and 4. We agree with the referee that the sources of these species are not unique and the main goal of this paper was to define the behavior of these species during specific events. The sources of each ion were described but it is not the main focus of this manuscript. Considering that MSA atmospheric concentration in NyA are dominated by biological bloom and sulphate and nitrate showed similar behaviour, the biogenic input was considered as the predominant source.

R: Line 329. Ions are not compounds.

A: As suggested by referee, we substituted "compounds" with "ion species"

R: Lines 328-360. Hard to understand. Descriptions on temporal changes in each ion do not seem to correctly correspond to Figures and Tables.

A: We corrected the numbers of figures and tables.

R: For example, although Lines 328-329 read that MSA showed rather homogeneous concentrations until the end of April, Table S1 shows very different values for three different depth intervals. There are other similar discrepancies between the text and Figures/Tables.

A: We modified the sentence (Lines 328-329) as follows: "The ion species related with the biogenic emission, in particular MSA, NO3-, and C5 had low concentrations without a specific stratification until the end of April." This is a general consideration about the behaviour of these species until April. We checked the concentration values reported for MSA but we confirmed those reported in manuscript.

R: The color plots in Figs. 2 and 3 are not easy to understand. Line graphs showing values are necessary here.

A: We thank the referee for the suggestion. We try to use line graphs, however the charts in the manuscript represent three variables, time (bottom axes), depth (y axes) and concentration (colors) and inspire from the chart used for the atmospheric particles load.

R: The authors argue that concentrations of biogenic ions increased in upper layers of the snowpack due to the input of primary production. Supporting evidence is necessary to show that the primary production did increase, air masses with high concentrations of biogenic species did reach AAB, and that these ions did penetrate into the snowpack is necessary.

A: To demonstrate the input increase of biogenic species in May, we inserted a reference of Park et al. (2018) where the authors described the "Atmospheric DMS in the Arctic Ocean and Its Relation to Phytoplankton Biomass" also in the spring 2015. I also wonder why sea spray species did not increase. If air masses from the open sea brought marine biogenic species to AAB, they could have also brought sea spray species. The reason to explain the difference is needed.

A: The referee affirms that "If air masses from the open sea brought marine biogenic species to AAB, they could have also brought sea spray species". The transport of ionic species depends to their particle size distribution in the atmospheric aerosol. The sea salt species are usually distributed in the coarse fraction of the aerosol and they have a local source because these coarse particles can deposit close to source. On the contrary, the biogenic species, such as MSA, are usually distributed in the fine fraction ($<1$ $\mu$m) of aerosol and the can undergo a long range atmospheric transport. The main difference between sea salt and biogenic species consists in the different particle distribution and so different source sites (Barbaro et al., 2019; Barbaro et al., 2017; Barbaro et al., 2017b).

R: Why were median concentrations, not averages, used?

A: We prefer to consider the median concentration in order to reduce the influence of

outliers.

R: The authors discuss increases and decreases. Are they statistically significant?

A: We used the statistical approach described (now with more details) in the section "Statistical Analysis" to establish the significance of the increases and decreases in the concentrations after the melting and rain events.

R: There should be spatial variability in ion concentrations in a snowpack. As concentrations were measured in different snow pits on different days, initial spatial variability should be taken into account to discuss temporal variability. I think authors can do that using the daily data from P1.

A: We adopted a sampling scheme to reduce the spatial variability. Snow pits were dug daily, perpendicular to the ice flow, and every new snow pit was dug approximately 30 cm upstream from the previous one. Although the results clearly show only the effects of melting and rain event, we cannot exclude a possible effect of spatial variability. To confirm this aspect, we added this sentence in the main manuscript and we introduce a new figure S2: "The spatial variability was evaluated by considering the total ion concentrations for each layers in five consecutive days relative to a period (7th – 11th April) without phenomena of snow, rain or wind that can modified the strata composition. The total ionic concentration (Fig. S2) showed the same profile in stable meteorological condition, suggesting that spatial variability was negligible compare to other processes that can modified the snow strata."

R: Line 333. Affected by what?

A: To clarify, we modified the sentence as follows: "The superficial layer (10-20 cm) was the stratum affected by the most evident variations of MSA and its concentrations varied from 16 ng g-1 to 34 ng g-1 in the P1 and P2, respectively, and 136 ng g-1 in the P3."

R: Lines 363-365. As mentioned above, this part needs more detailed explanation.
A: We are sorry for the quality of presentation. We revise the text of the discussion to make clear that the results are based on the methods described in the section entitled "Statistical Analysis".

R: Line 369. Where were the values 15.1mm and 15mm recorded? The values would be different at the sampling site.

A: We reported the details about the meteorological station in the "Methods" section as follows: "... the daily precipitation data were recorded in Ny-Ålesund by the Norwegian Meteorological Institute (station n. 99910) and downloaded through the eKlima database (eklima.no)." From the observatory at on Mt. Zeppelin (http://ebas.nilu.no/Pages/Plot.aspx?key=DBCBA03A77D54265868218D4E5E63521) is possible to obtain temperature data but the precipitation data are not available. We had to refer to the data measured at NyA.

R: Lines 387 and 391. Line 387 reads that C5 and C2 are biogenic organic ions, but line 391 reads that these ions originated from continental pollution sources. This is confusing.

A: The source attribution of these compounds is also debated but we consider these species biogenic because they had the same behavior of MSA. These diacids are directly emitted to the atmosphere by fossil fuel combustion and biomass burning and are produced in the atmosphere by secondary photochemical oxidations of anthropogenic and also natural organic compounds (Kawamura and Sakaguchi, 1999). We agree with referee that this aspect is quite confuse and for this reason we added reference and we modified the manuscript as follows: "Biogenic organic ions (C5 and C2) showed a weakly increase of concentration in the upper part of snowpack, due to deposition of secondary aerosols (Fig. 3). These diacids are directly emitted to the atmosphere by fossil fuel combustion and biomass burning and are produced in the atmosphere by secondary photochemical oxidations of anthropogenic and natural organic compounds (Kawamura and Sakaguchi, 1999). They can be long-range transported in the atmosphere because they are mainly distributed in the fine aerosol particles (Kawamura et al., 2007)."

R: Line 400. Does "temperate" mean "temperature"?

A: Thanks. We corrected it.

R: Lines 405-406. What does "an improve of concentrations" mean?

A: We modified "improve" with "increase"

R: Lines 407-420. As stated above, I couldn't understand how the elution sequences were derived.

R: Lines 421-434. Unless ion concentrations in the rain and fresh snow deposited on the snow surface are known, I think it is difficult to discuss the causes of the changes in ion concentrations.

A: We partially agree with the referee since we cannot estimate the concentration in the rain during the 16th of April. However, we can estimate the concentration of the fresh snow fall since we sampled them in the upper 10 cm. We are interested in what remains on the surface of the snow pack and not in the concentration in the deposition. This is another interesting topic but beyond the focus of the manuscript. For example, we suggest in the manuscript that iodine deposit mainly during the snow fall events since every time there is a snow accumulation we note an increase of its concentration on the surface layer. In addition, the concentration in the snow deposit on the ground changes with the time, with likely higher concentrations in first snow flakes deposit due to initial higher atmospheric load, and lower in the later atmospheric load. Considering the wind effect is particularly difficult to associate which layers are maintained on the surface of the snow pack and which one is removed, brought to an over estimation of the data. To evaluate this point, a specific study and protocol should be adopted, as suggested by the referee, with a higher temporal resolution. We focus our study only in what's happen in the snow pack and the effect of melting events.

R: Lines 440-442. I couldn't understand English here.

A: We tried to clarify the sentence as follows: ": during the rain event, ion species undergo a vertical remobilization while the effect of ion mobilization produced by the melting period is horizontally distributed and uniform in the whole snow pack."

R: Line 453. Fig.5?

A: We checked all figure numbers and we corrected the mistakes.

R: Line 480. How is "From a millennial scale" relevant to this study?

A: We removed it.

R: Was the density of each snow layer measured? Density measurement is essential for mass balance calculations.

A: The density was measured ones and ranged between 0.35 to 0.45 (using 10 cm resolution) except the fresh snow fall. In our study we are measuring concentration and not flux where the density measurements are necessary. The aim of the paper is to identify a change in concentration for a specific depth (considering the correction adopted for the snow gain\loss) and not the flux for each snow layer, giving indication of the depositional flux for a specific events.

R: Fig. 1. First, figure title, a, b, and c are missing.

A: Sorry, but the figure title is reported in the figure and in the caption.

R: Second, the color plots are not easy to understand. In the uppermost plot, value zero is very important, but difficult to see. In the lowermost plot, the melting point is very important, but difficult to see. Line plots showing changes in values are needed here. Third, the manuscript needs to explain how the lowermost plot was made from the 11 temperature probes.

A: As suggested by referee#2, we modified the figure 1 to better show the difference in
the temperature. As suggested by referee#1, we have also modified the air temperature plot with a line plot.

R: Fig. S1. Visual stratigraphy should be presented. At least new snow and ice layers should be marked.

A: As also suggested by referee#1, we inserted the Figure 2 in the main manuscript with the evolution of the snow stratigraphy in the first meter of annual snow. All data are corrected in function of the accumulation. The layers detected are classified by their resistance of penetration (hardness) using the hand test. The hand test identifies very soft snow with the number 1 while hard snow was described by the number 5.

R: There are also other typos and errors that I haven't pointed out.

A: We checked all manuscript to remove the errors.